# Nucleolar dynamics and interactions with nucleoplasm in living cells

Christina M Caragine, Shannon C Haley, Alexandra Zidovska*

Center for Soft Matter Research, Department of Physics, New York University, New York, United States

**Abstract** Liquid-liquid phase separation (LLPS) has been recognized as one of the key cellular organizing principles and was shown to be responsible for formation of membrane-less organelles such as nucleoli. Although nucleoli were found to behave like liquid droplets, many ramifications of LLPS including nucleolar dynamics and interactions with the surrounding liquid remain to be revealed. Here, we study the motion of human nucleoli *in vivo*, while monitoring the shape of the nucleolus-nucleoplasm interface. We reveal two types of nucleolar pair dynamics: an unexpected correlated motion prior to coalescence and an independent motion otherwise. This surprising kinetics leads to a nucleolar volume distribution, $p(V) \sim V^{-1}$, unaccounted for by any current theory. Moreover, we find that nucleolus-nucleoplasm interface is maintained by ATP-dependent processes and susceptible to changes in chromatin transcription and packing. Our results extend and enrich the LLPS framework by showing the impact of the surrounding nucleoplasm on nucleoli in living cells.

## Introduction

The nucleolus is the largest structure present in the cell nucleus of eukaryotic cells. This membrane-less organelle is a site of ribosomal biogenesis and plays a key role in cell cycle progression and stress response (*Alberts et al., 2014*; *Montanaro et al., 2008*; *Boulon et al., 2010*). Nucleoli are composed of RNA and proteins and embedded in the chromatin solution inside the nucleus. They form at specific parts of genome called nucleolar organizer regions (NORs) containing rDNA, which is transcribed inside the nucleolus (*McClintock, 1934*; *Ritossa and Spiegelman, 1965*; *Wallace and Birnstiel, 1966*). At the beginning of the cell cycle a small nucleolus forms at each NOR. These nucleoli later fuse into larger ones, while remaining connected to their NORs in somatic cells (*Amenta, 1961*; *Sullivan et al., 2001*).

The lack of a nucleolar membrane has long been intriguing biologists and physicists alike, questioning the physical nature of the nucleolus. Pioneering studies in frogs found that nucleoli in *X. laevis* oocytes behave like liquid droplets *in vivo*, as well as when reconstituted *in vitro*, and suggested that nucleoli form through liquid-liquid phase separation of the nucleolar components in the nucleoplasm (*Brangwynne et al., 2011*; *Berry et al., 2015*; *Feric et al., 2016*). The volume distribution of such nucleoli was in agreement with a diffusion-limited aggregation process with a constant influx of particles (*Brangwynne et al., 2011*). In addition, the size of nucleoli in the worm *C. elegans* embryos was found to be dependent on the concentration of nucleolar components in the nucleoplasm which is consistent with the liquid-like nature of the nucleolus (*Weber and Brangwynne, 2015*). The nucleolar subcompartments, that is the granular and the dense fibrillar components, were also suggested to form via liquid-liquid phase separation (*Feric et al., 2016*). Recent studies in the fly *D. melanogaster* suggest that while the nucleolar assembly follows liquid-liquid phase separation, active protein recruitment is also involved (*Falahati and Wieschaus, 2017*).

Recently, we have shown that human nucleoli also exhibit liquid-like behavior (*Caragine et al., 2018*). By analyzing the shape fluctuations of nucleolar surface and kinetics of the nucleolar fusion in

*For correspondence:
alexandra.zidovska@nyu.edu

Competing interests: The authors declare that no competing interests exist.

**eLife digest** The inside of a cell is very organized. Just as bodies contain internal organs, cells contain many different compartments, called 'organelles', each with its own specific role. Most organelles are surrounded by a membrane that keeps their contents separate from the cytoplasm, the water-based liquid inside the rest of the cell.

Some organelles, however, are not bounded by a membrane. Instead, they act like tiny drops of oil in water, retaining their structure because they have different physical properties from the fluid around them, a phenomenon called liquid-liquid phase separation.

One such organelle is the nucleolus, which sits inside the cell's nucleus (a membrane-bound organelle containing all the genetic material of the cell in the form of DNA). The nucleolus's job is to produce ribosomes, the cellular machines that, once transported out of the nucleus, will make proteins.

Human cells start with 10 small nucleoli in the nucleus, which fuse together until only one or two larger ones remain. Previous research showed that nucleoli form and persist thanks to liquid-liquid phase separation, and they behave like liquid droplets. Despite this, exactly how nucleoli interact with each other and with the fluid environment in the rest of the nucleus remained unknown. Caragine et al. set out to measure the behavior and interactions of nucleoli in living human cells.

Microscopy experiments using human cells grown in the laboratory allowed the positions, size and shape of nucleoli to be tracked over time. This also yielded detailed information about the smoothness of their surface. Mathematical analysis revealed that pairs of nucleoli normally moved independently of each other, unless they were about to fuse, when they invariably slowed down and coordinated their movements. In addition, altering the state of DNA in the surrounding nucleus also affected the nucleoli. For example, when DNA was less densely packed, nucleoli shrank and their surfaces became smoother.

These results build on our knowledge of how cells are organized by showing, for the first time, that the environment within the nucleus directly shapes the behavior of nucleoli. In the future, a better understanding of how cells maintain healthy nucleoli may help develop new treatments for human diseases such as cancer, which are characterized by problems with this organelle.

human cells *in vivo*, we found nucleolar dynamics to be consistent with that of liquid droplets with very low surface tension $\gamma \sim 10^{-6}$ Nm$^{-1}$ surrounded by highly viscous nucleoplasm of viscosity $\eta \sim 10^3$ Pa s (*Caragine et al., 2018*). Strikingly, it is the nucleoplasm viscosity that sets the time scale for the nucleolar coalescence providing resistance to the already very low surface tension that drives the process (*Caragine et al., 2018*). Correspondingly, nucleolar coalescence in human cells takes hours to complete (until the newly formed nucleolus rounds up, *Figure 1A*), while the neck connecting two coalescing nucleoli is discernable only for minutes after their initial touch (*Figure 1B*) and its radius $r$ grows in time as $r(t) \sim t^{1/2}$ (*Caragine et al., 2018*). Such long coalescence times have been speculated not to interfere with the rDNA transcription inside the nucleoli (*Caragine et al., 2018*).

The nucleoplasm (chromatin solution) and its physical properties clearly contribute to the nucleolar physiology. Interestingly, while the nucleolar coalescence can be described by a theory of passive liquid droplets within a highly viscous passive fluid (*Caragine et al., 2018*; *Paulsen et al., 2014*), nucleoplasm is an active fluid. Specifically, chromatin dynamics was shown to be active, that is ATP-dependent, and coherent, that is exhibiting correlated displacements, over 3–5 μm in human cells (*Zidovska et al., 2013*). Thus, the measured $\gamma$ and $\eta$ are likely effective quantities (*Caragine et al., 2018*). Chromatin is known to localize as a denser heterochromatin at the nucleolar surface (*Padeken and Heun, 2014*), yet the nature of physical interactions between the nucleolar surface and the chromatin solution remains to be revealed (*Németh and Längst, 2011*; *Bickmore and van Steensel, 2013*; *Towbin et al., 2013*). Disruption and dysfunction of the nucleolus is implicated in a large number of human diseases, such as skeletal and neurodegenerative disorders, cardiovascular disease and cancer (*Hannan et al., 2013*; *Núñez Villacís et al., 2018*; *Ruggero and Pandolfi, 2003*; *Derenzini et al., 2009*). Thus, elucidating physical principles governing the nucleolus-nucleoplasm interface might contribute to our understanding of the nucleolus in health and disease.

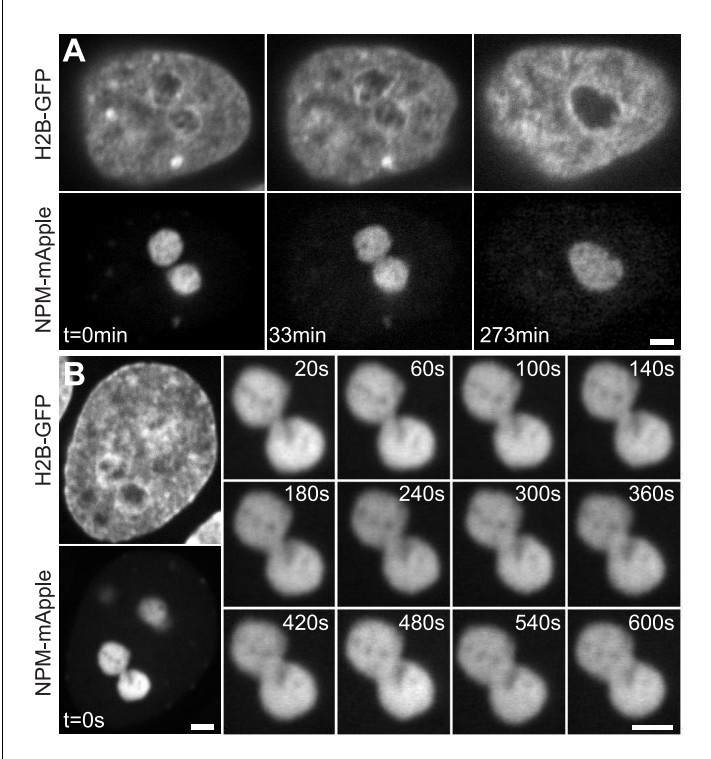

**Figure 1.** Nucleolar coalescence. (**A**) Time lapse of a nucleus with fluorescently labeled chromatin (H2B-GFP) and two fusing nucleoli (NPM-mApple). Time points depict: *pre-fusion* ($t = 0$ min), with two distinct nucleoli, *fusion in progress* ($t = 33$ min), with a clearly visible neck connecting the two nucleoli, and *post-fusion* ($t = 273$ min), where a resultant nucleolus can be seen still rounding up. (**B**) Time series showing the growth of the neck connecting two coalescing nucleoli. At $t = 0$ s, both the fluorescently labeled chromatin (H2B-GFP) and coalescing nucleoli (NPM-mApple) are depicted, the later frames, 20–600 s, show the progress of the nucleolar coalescence (NPM-mApple). Parts of (**B**) adapted from Figure 3b in *Caragine et al. (2018)*. Scale bar, 2 μm.

In this work, we investigate the physical interactions between the nucleoli and the surrounding nucleoplasm by studying the structural features and dynamical behavior of the nucleoli. Specifically, to illuminate the kinetics of nucleolar assembly process, we examine the changes in the nucleolar size distribution with progressing cell cycle. In addition, we probe the physical nature of the nucleolar subcompartments, specifically, the granular components and the dense fibrillar components, and their contribution to the nucleolar liquid-like properties. To elucidate the role of nucleoplasm in nucleolar coalescence, we interrogate size, shape, position and alignment, as well as mobility inside the nucleus for both nucleoli that are about to fuse as well as those that do not fuse. To determine the role of active processes in maintaining the liquid-like nucleolus-nucleoplasm interface, we deplete ATP and further evaluate its structure and dynamics. Finally, we probe the contribution of specific cellular processes (such as cytoskeletal forces, transcriptional activity as well as protein synthesis) to maintaining the nucleolus-nucleoplasm interface by employing targeted biochemical perturbations.

## Results

### Nucleolar size distribution during the cell cycle

To address the kinetics of the nucleolar assembly process, we have evaluated the number and size of the nucleoli at different times during the cell cycle. After mitosis, human nuclei initially contain 10 nucleoli, which later fuse to form fewer larger ones (*Savino et al., 2001*). Thus, due to the changing nucleolar number, the likelihood of their coalescence is expected to vary with the cell cycle

progression. First, we measure the nucleolar size distribution in an unsynchronized cell population, which contains cells at all cell cycle stages (*Figure 2A*). Then we obtain the specific nucleolar size distributions at different, well-defined times of the cell cycle by synchronizing the cell population and monitoring their nucleolar count and size with progressing cell cycle (*Figure 2B–D*). Specifically, we carry out our measurements 1.5 hr and 3 hr after mitosis as well as at the end of the cell cycle, at the G2/M check point (*Figure 2B–D*). At every time point, we collect data from the entire volume of the cell nucleus by taking a z-stack with focal planes 0.5 μm apart. *Figure 2A–D* shows micrographs of nuclei with fluorescently labeled chromatin (H2B-GFP) and nucleoli (NPM-DsRed) for all studied populations, respectively. Moreover, *Figure 2*, insets 1–4, shows an enlarged view of the boxed in nucleus from *Figure 2A–D*, respectively. While *Figure 2*, inset 1 depicts a nucleus from an unsynchronized cell population, *Figure 2*, insets 2–3 show the same nucleus with progressing time. Note, the presence of both small and large nucleoli early in the cell cycle (*Figure 2*, inset 2–3), with the

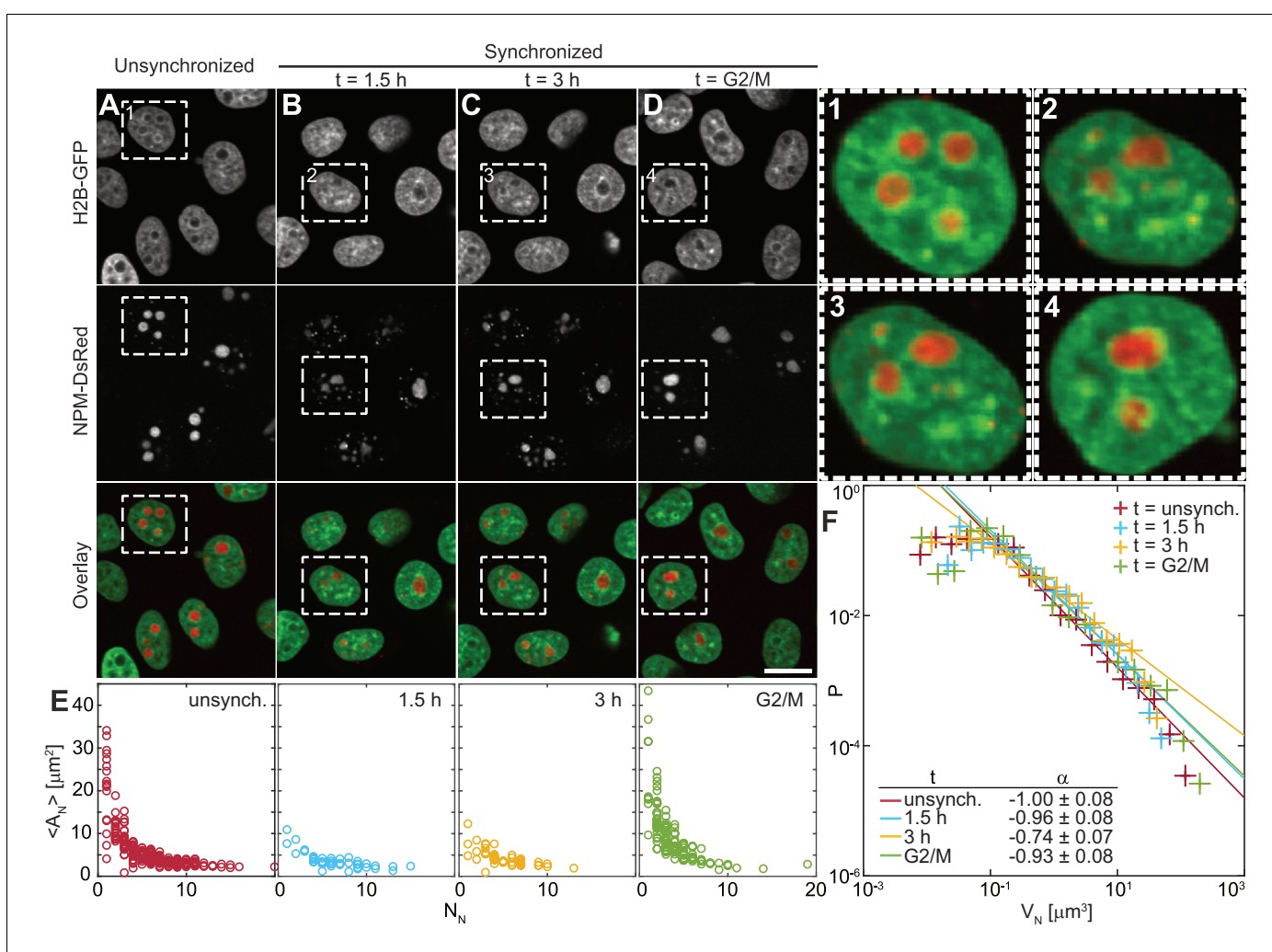

**Figure 2.** Nucleolar size distribution as a function of cell cycle. (A–D) Micrographs of HeLa cell nuclei with fluorescently labeled chromatin (H2B-GFP) and nucleoli (NPM-DsRed) under the following conditions: unsynchronized cells (A), synchronized cells 1.5 hr (B) and 3 hr (C) after mitosis, and cells arrested at G2/M checkpoint (D). (1–4) enlarged view of the boxed nuclei from (A–D). (E) Average nucleolar area $A_N$ as a function of number of nucleoli per nucleus $N_N$ for all conditions from (A–D). For unsynchronized cells, total number of nucleoli analyzed $N_N$ = 1331 in 228 nuclei, for $t$ = 1.5 hr, $N_N$ = 275 in 42 nuclei, for $t$ = 3 hr, $N_N$ = 257 in 51 nuclei, and for $t$ = G2/M, $N_N$ = 497 in 124 nuclei. (F) Distributions of nucleolar volume $V_N$ and their fit to $f(V_N) \sim V_N^\alpha$ for all conditions from (A–D). For all fits, the goodness-of-fit, $R^2 > 0.98$. Scale bar, 15 μm.
The online version of this article includes the following figure supplement(s) for figure 2:

**Figure supplement 1.** Nucleolar area distributions, $p(A_N)$, for all conditions shown in *Figure 2E*: unsynchronized cells, synchronized cells 1.5 hr after mitosis, synchronized cells 3 hr after mitosis, and cells arrested in G2/M.

large ones becoming more spherical between 1.5 hr and 3 hr after mitosis, while only large nucleoli are seen at the end of the cell cycle (*Figure 2*, inset 4).

*Figure 2E* shows the distributions of average nucleolar area of nucleoli in one nucleus, $\langle A_N \rangle$, as a function of the nucleolar number in the given nucleus, $N_N$, for the unsynchronized and synchronized cell populations at the studied time points. The distributions of nucleolar area, $A_N$, for each time point are shown in *Figure 2—figure supplement 1*. For each nucleolus we measure its area in its respective focal plane within the collected z-stack. We find that as the cell cycle progresses, the number of nucleoli per nucleus decreases, while the average nucleolar area in the nucleus increases (*Figure 2E*). Interestingly, this trend persists beyond 3 hr into the cell cycle suggesting that the fusion of nucleoli is not limited to the first two hours of the cell cycle as previously hypothesized (*Savino et al., 2001*). To gain further mechanical insight into the nucleolar coalescence kinetics during the cell cycle, we have analyzed the nucleolar volume distribution for each time point (*Figure 2F*). We calculated nucleolar volume assuming a spherical shape, $V_N = 4\pi r^3/3$, where $r$ is the radius of a circle with the area equal to the nucleolar area, and using the least square method we fitted the nucleolar volume distribution $P(V_N)$ to a power law $f(V_N) \sim V_N^\alpha$. Our data shows that $P(V_N)$ can be described by a power law with $\alpha \sim -1$ for all cell populations, unsynchronized as well as synchronized at all studied time points. The confidence intervals for the fitting parameter $\alpha$ are listed in *Figure 2F* with the goodness-of-fit $R^2 > 0.98$ for all fits. It is noteworthy, that such distribution is divergent, and so is its first moment, the mean, if integrated over all volumes (from 0 to $\infty$). However, the measured $p(V)$ distribution does have finite bounds given by the physical cut-offs for the nucleolar size, the minimum and maximum that it can reach inside a cell nucleus.

## Physical nature of nucleolar subcompartments

The human nucleolus behaves like liquid droplet (*Caragine et al., 2018*), yet the nucleolar fluid is complex, containing three distinct subcompartments; fibrillar center (FC), dense fibrillar component (DFC) and granular component (GC). They all play a different role in ribosome biogenesis and vary in protein composition: While FC contains polymerase I, DFC and GC are enriched in fibrillarin (FBL) and nucleophosmin (NPM), respectively (*Boisvert et al., 2007*). Moreover, they show a hierarchical organization, suggested to form via liquid-liquid phase separation (*Feric et al., 2016*), with FCs nested inside DFCs, which are embedded in GC.

To address the contributions of these subcompartments to the overall liquidity of the human nucleolus, we examine their physical properties. *Figure 3A* shows micrographs of three different nuclei with fluorescently labeled chromatin (H2B-GFP, green), GC (NPM-DsRed, red) and DFCs (FBL-mCerulean, blue). We obtain the nuclear and nucleolar contours from H2B-GFP and NPM-DsRed signal, respectively. By analyzing the FBL-mCerulean signal we procure the shape, size and number of DFCs inside a nucleolus. A visual inspection of our data reveals that DFCs appear to be close to spherical. To verify this observation, we measure the DFC eccentricity: First, we measure the length of the semi-major DFC axis $a$ and the semi-minor DFC axis $b$ (Materials and methods). *Figure 3B* displays the distributions of measured lengths of both $a$ (red) and $b$ (green), together with the Gaussian fits $f(a_{DFC}) \sim e^{(a_{DFC} - \langle a_{DFC} \rangle)^2 / 2\sigma_{a_{DFC}}^2}$ (red line) and $f(b_{DFC}) \sim e^{(b_{DFC} - \langle b_{DFC} \rangle)^2 / 2\sigma_{b_{DFC}}^2}$ (green line) of their respective distributions. From the Gaussian fits we obtain the following average values: $\langle a_{DFC} \rangle = 210 \pm 50$ nm and $\langle b_{DFC} \rangle = 180 \pm 40$ nm. Next, we evaluate the eccentricity $e = a/b$ for each DFC and find that the DFC shape is indeed close to spherical with the average eccentricity $\langle e \rangle = 1.22 \pm 0.17$ (where $e = 1$ corresponds to a circle) and average area of $\langle A_{DFC} \rangle = 0.13 \pm 0.06$ μm², where $A_{DFC} = \pi a b$. The distributions of $e$ and $A_{DFC}$ are shown in *Figure 3—figure supplement 1*. Overall, we identified 1279 DFCs over 114 nucleoli in 63 nuclei, and after the removal of the DFCs that were out of focus, we obtain measurements of $a$, $b$, $e$, and $A_{DFC}$ for 1035 DFCs.

Next, we evaluate the nucleolar area, $A_N$, as a function of the DFC number, $N_{DFC}$, inside the given nucleolus (*Figure 3C*). Our data reveals that $A_N$ grows linearly with $N_{DFC}$, with a linear fit of $A_N = (0.92 \pm 0.05)N_{DFC}$. This implies that upon nucleolar coalescence, which leads to larger $A_N$, the new nucleolus contains the cumulative number of DFCs, indicating that DFCs do not fuse themselves. This is further corroborated by the volume distribution of DFCs, $p(V_{DFC})$, which has a sharp peak at $V_{DFC} = 0.03$ μm³ (*Figure 3—figure supplement 1*), indicating that DFCs are largely monodisperse. Moreover, this finding suggests that every DFC is associated with a GC domain of an area $A_{GC} \approx 0.79$ μm². Since we found DFCs to exhibit a close to spherical shape, we can estimate the

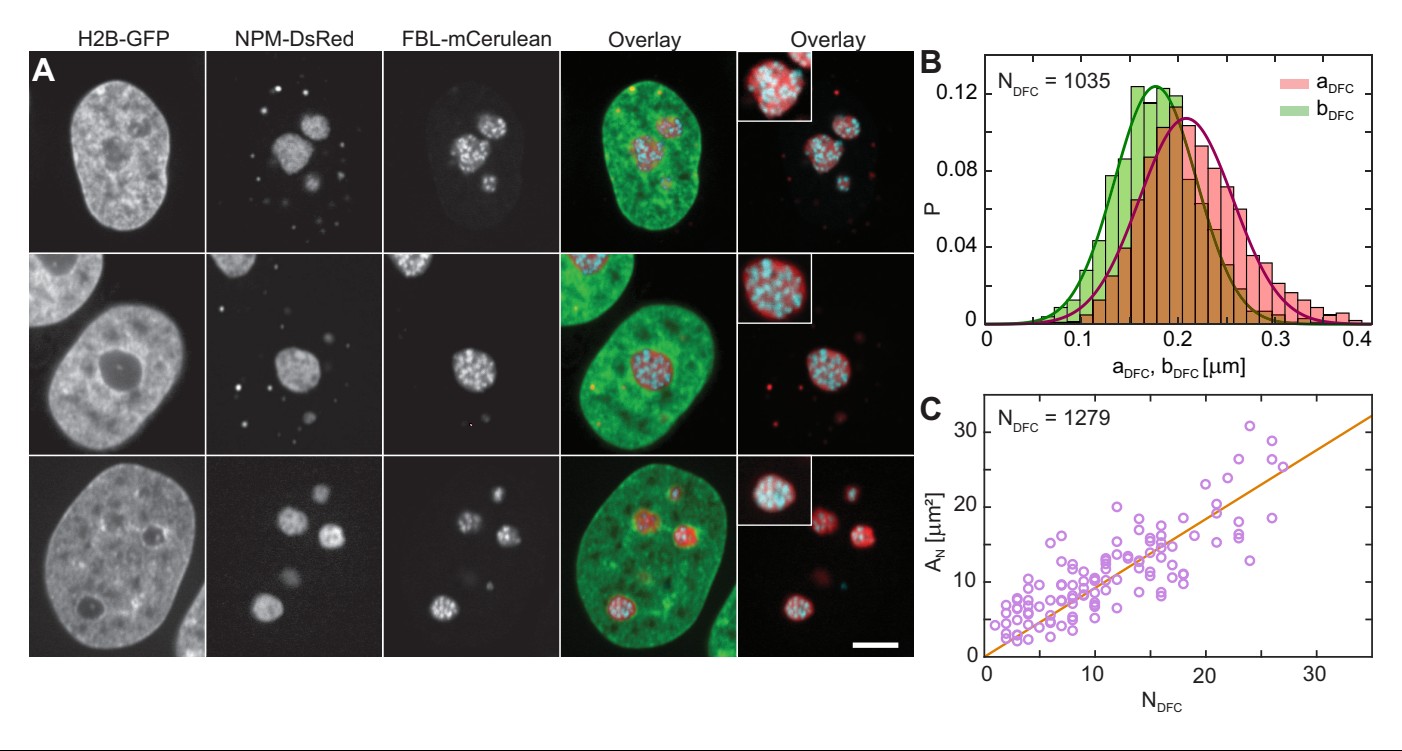

**Figure 3.** Nucleolar internal structure. (**A**) Micrographs of HeLa nuclei with fluorescently labeled chromatin (H2B-GFP, green), nucleolar granular component (NPM-DsRed, red) and nucleolar dense fibrillar component (DFC) (FBL-mCerulean, blue) and overlays of all three signals (green, red, blue) and red and blue signal. The insets in overlay images present an enlarged view of a nucleolus from the image. (**B**) Distributions measured for semi-major axis $a_{DFC}$ (red) and semi-minor axis $b_{DFC}$ (green) of single DFCs ($N_{DFC}$ = 1035). The solid red and green lines correspond to the Gaussian fits of distributions of $a_{DFC}$ and $b_{DFC}$, respectively, with $\langle a_{DFC}\rangle \approx$ 210 nm and $\langle b_{DFC}\rangle \approx$ 180 nm. (**C**) Nucleolar area, $A_N$, as a function of DFC number per nucleolus, $N_{DFC}$, with a linear fit $A_N \approx 0.92 N_{DFC}$. We evaluated 1279 DFCs over 114 nucleoli in 63 nuclei. Scale bar, 5 μm.

The online version of this article includes the following figure supplement(s) for figure 3:

**Figure supplement 1.** Distributions of DFC eccentricity, area and volume.

volume fraction of DFCs and GC phase in the human nucleolus, and find $\Phi_{DFC} \approx 0.1$ and $\Phi_{GC} \approx 0.9$, respectively.

## Fusing and non-fusing nucleoli

Our recent study revealed that the timescale of the nucleolar coalescence is set by the high viscosity of the surrounding nucleoplasm ($\eta_{np} \sim 10^3$ Pa s) (*Caragine et al., 2018*). To elucidate the physical interactions of nucleolar droplets with the chromatin solution, we interrogate their size, shape, position and alignment inside the cell nucleus. Moreover, we compare these characteristics for nucleoli that fuse and the ones that do not fuse during our observation. For non-fusing nucleoli, we record time lapses for 60 min with a time step of 5 min and at every time step we collect a z-stack with focal planes 0.5 μm apart. By collecting a z-stack, we can monitor all nucleoli present in the given nucleus and obtain measurements in their respective focal planes. To capture a fusion of nucleoli, we observe a pair of nucleoli for 60–270 min with a time step of 5–15 min, and review at the end of the experiment if the fusion has occurred. In three cases, we were able to track three or four nucleoli simultaneously, the nucleoli closest together were then defined as pairs. In case of three nucleoli only one pair was analyzed, that is two closest nucleoli. In one case, a nucleolar pair fused while the measurement was being set up, and was therefore only analyzed in the post-fusion nucleolar population.

*Figure 4A* shows micrographs of a nucleus with fluorescently labeled chromatin (H2B-GFP) at $t$ = 0 and 60 min, where the nucleoli correspond to the voids in the H2B-GFP signal and are highlighted by symbols (*circle* and *triangle*). In contrast, *Figure 4D* shows micrographs of nucleus with

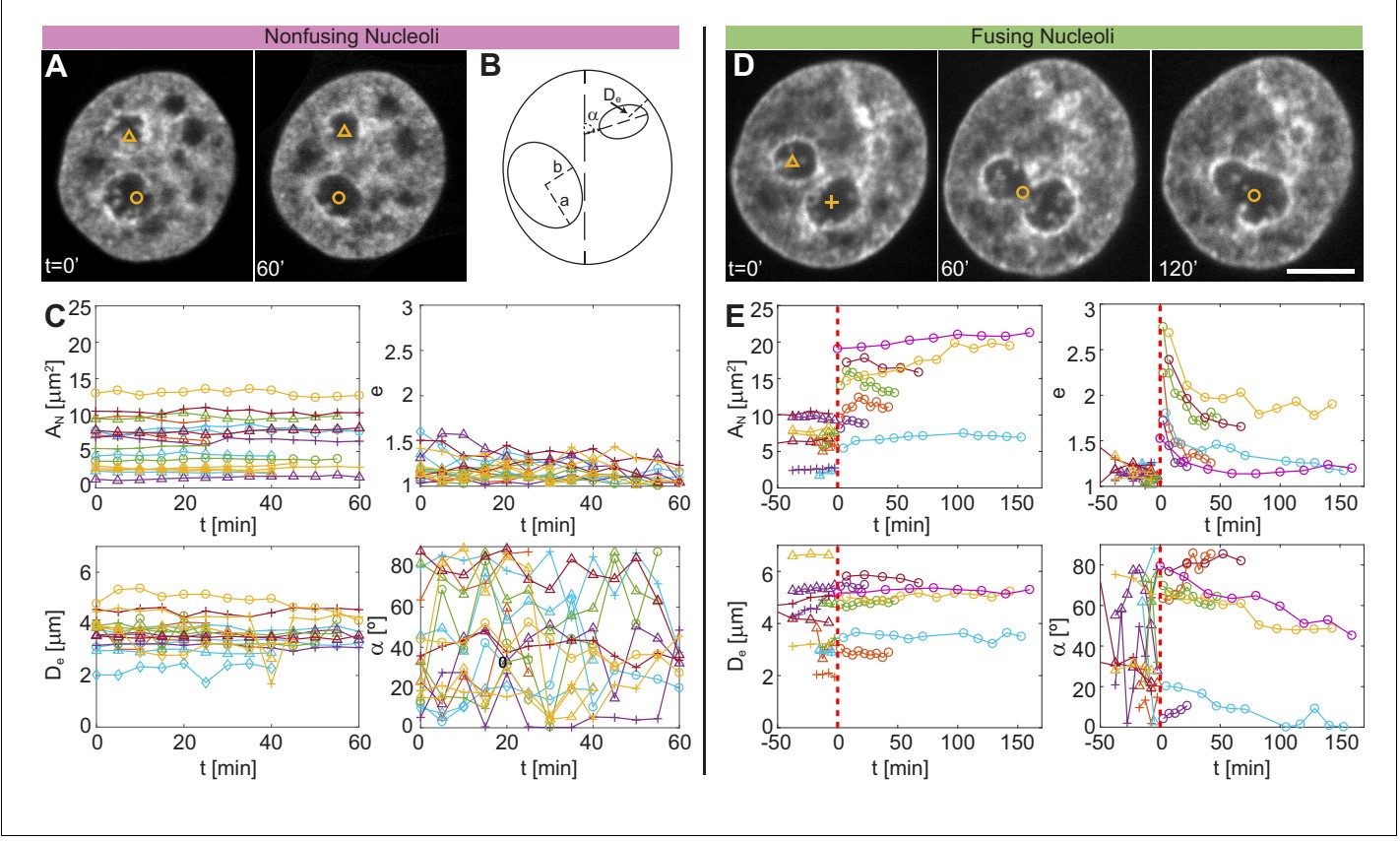

**Figure 4.** Comparison of size, shape and nuclear positioning between fusing and nonfusing nucleoli. (A) Micrographs of a nucleus with fluorescently labeled chromatin (H2B-GFP), where two void spaces (labeled by *yellow triangle* and *yellow circle*) correspond to two nucleoli that did not fuse between $t = 0$ and 60 min. (B) Schematics of measured variables. (C) Measured variables for nonfusing nucleoli: nucleolar area, $A_N$, nucleolar eccentricity, $e$, shortest distance from the nucleolar centroid to the nuclear envelope, $D_e$, and the angle between the major nuclear and nucleolar axes, $\alpha$ ($N_N = 17$, $N_{cell} = 6$). All characteristics are calculated in the nucleolar focal plane. (D) Micrographs of a nucleus with fluorescently labeled chromatin (H2B-GFP), where two void spaces (labeled by *yellow triangle* and *yellow cross*) correspond to two nucleoli before fusion at $t = 0$ min, while at $t = 60$ and 120 min they can be seen fusing (*yellow circle*). (E) Measured variables for fusing nucleoli: $A_N$, $e$, $D_e$ and $\alpha$ ($N_N = 12$, $N_{cell} = 7$). The dashed red line at $t = 0$ min indicates fusion. All measurements are carried out in the nucleolar focal plane. Scale bar, 5 µm.

fluorescently labeled chromatin (H2B-GFP) at $t = 0$, 60 and 120 min, with fusion occurring shortly before $t = 60$ min. The nucleoli correspond to the voids in the H2B-GFP signal and are highlighted by *triangle* and *cross* before fusion and by *circle* during and after fusion.

Next, we obtain contours for all nucleoli in their respective z-plane and measure their area, $A_N$, by filling their contour. To evaluate the nucleolar shape we compute its eccentricity, $e = a/b$, with $a$ and $b$ being the semi-major and semi-minor axes of a fitted ellipse, respectively. For $e = 1$, the nucleolus is spherical, while for $e > 1$ the nucleolus has an elliptical shape. Further, we determine the shortest distance of the nucleolar centroid to the nuclear envelope, $D_e$, as well as the angle between the nuclear and nucleolar major axes, $\alpha$, when fitted by an ellipse, respectively. *Figure 4B* provides an illustration of the measured parameters $a$, $b$, $D_e$ and $\alpha$. First, we evaluate these quantities for the non-fusing nucleoli (*Figure 4C*). We find that $A_N$, $e$ and $D_e$ do not change appreciably, while $\alpha$ fluctuates significantly during the duration of the experiment. In fact, a constant area might indicate that there is no significant addition or removal of nucleolar material during this time. The eccentricity is rather low, often close to 1, making $\alpha$ susceptible to small fluctuations.

For comparison, *Figure 4E* shows the same quantities for the nucleoli that fused during the experiment. We aligned the different fusion events in time with fusion occurring at $t = 0$ min, as marked by the red dashed line. The pairs of fusing nucleoli are identified by symbols of the same color, with two nucleoli before fusion indicated by a *triangle* and *cross*, and the new nucleolus after fusion by a *circle*. Interestingly, $A_N$ upon fusion is about same as the summed area of both pre-fusion

nucleoli. Since we do not observe any significant increase of $A_N$ after fusion, there is likely no significant material influx associated with nucleolar fusion. As expected, $e$ decreases after fusion for all nucleoli, consistent with our prior finding of surface tension driving the fusion (*Caragine et al., 2018*). $D_e$ does not show any leading trends for the position of the new nucleolus; some remain closer to a position of one of the pre-fusion nucleoli, whereas some move into an intermediate $D_e$ of the two pre-fusion nucleoli. Interestingly, our $\alpha$ measurement shows that nucleoli after fusion are slowly moving into a parallel ($\alpha = 0°$) or a perpendicular ($\alpha = 90°$) alignment with the major nuclear axis.

## Dynamics of fusing and non-fusing nucleoli

To further elucidate the nucleolus-nucleoplasm interactions, we investigate the dynamics of both non-fusing and fusing nucleoli. We track the nucleolar motion, by tracking its centroid in time, and obtain a trajectory for every nucleolus. By analyzing nucleolar trajectories, we can extract the nucleolar velocity, $v$, which we measure relative to the nuclear centroid. Further, we evaluate the radial velocity, $v_{rad}$, which we define as the nucleolar velocity along the line connecting centroids of two nucleoli, with origin being the line center. $v_{rad}$ allows us to assess the motion of one nucleolus towards another, where a negative value of $v_{rad}$ corresponds to motion towards the other nucleolus. We also compute the angle, $\alpha_v$, between the nucleolar velocity $v$ and the line connecting the two nucleoli, with $\alpha_v$ ranging from 0° to 180°, informing if a nucleolus is traveling towards the other.

*Figure 5A* shows examples of trajectories for non-fusing nucleoli, with their temporal evolution color-coded (blue to red). An enlarged view of these trajectories and the areas they cover is depicted in *Figure 5—figure supplement 1A–B*. A distribution of nucleolar velocities $v$ for non-fusing nucleoli is shown in *Figure 5B*, with a mean velocity $\langle v \rangle \approx (0.49 \pm 0.30) \times 10^{-3}$ μms$^{-1}$. Next, we look at the dynamic behavior of a nucleolar pair, and analyze their velocities with respect to each other. We plot the larger velocity, $v_{max}$, against the smaller velocity, $v_{min}$ (*Figure 5C*). The scatter plot in *Figure 5C* shows a wide spread and no apparent correlation between $v_{max}$ and $v_{min}$ (Pearson correlation coefficient $\rho = 0.41$). Moreover, a distribution of $v_{rad}$ is centered around 0, thus not pointing towards any preferred direction of motion (*Figure 5D*). This observation is further corroborated, when we find $v_{rad}$ to fluctuate around 0 as a function of time (*Figure 5E*), as well as $\alpha_v$ changing seemingly randomly with time (*Figure 5F*). By fitting a Gaussian curve to the $v_{rad}$ distribution in *Figure 5D* we obtain a variance $\sigma_{rad,nonfusing} = 3 \times 10^{-4}$ μms$^{-1}$.

Next, we analyze the motion of pairs of fusing nucleoli in the same fashion. Examples of such trajectories, with their temporal evolution color-coded (blue to red), are shown in *Figure 5G*. An enlarged view of these trajectories and the areas they cover is depicted in *Figure 5—figure supplement 1C–D*. *Figure 5H* shows the distribution of the nucleolar velocities, $v$, for fusing nucleoli, with an average velocity $\langle v \rangle \approx (0.33 \pm 0.26) \times 10^{-3}$ μms$^{-1}$, which is ~ 30% smaller than for non-fusing nucleoli. This difference is statistically significant as corroborated by $p$-value of $4 \times 10^{-4}$. Strikingly, when we review velocities of a fusing nucleolar pair and plot the larger velocity, $v_{max}$, against the smaller velocity, $v_{min}$ (*Figure 5I*), we find a clear linear correlation between them, with the linear fit of $v_{max} = (1.74 \pm 0.20)v_{min}$ (*Figure 5I*, blue line) and a Pearson correlation coefficient $\rho = 0.88$. Moreover, we find that $\langle v_{max}/v_{min} \rangle \approx 1.8 \pm 0.6$. The distribution of $v_{rad}$ for fusing nucleoli (*Figure 5J*) is still centered around 0, but is clearly narrower than for non-fusing nucleoli (*Figure 5D*) with a variance $\sigma_{rad,fusing} = 1 \times 10^{-4}$ μms$^{-1}$ obtained by fitting a Gaussian curve to the $v_{rad}$ distribution in *Figure 5J*. $\sigma_{rad,fusing}$ is about three times smaller than $\sigma_{rad,nonfusing}$. When reviewed over time, $v_{rad}$ exhibits much smaller fluctuations (*Figure 5K*) than in case of non-fusing nucleoli (*Figure 5E*). Lastly, *Figure 5L* shows $\alpha_v$ of the pre-fusion nucleoli as a function of time, monitoring 40 min prior to fusion, which occurs at $t = 0$ min. Interestingly, the nucleoli seem not to follow any preferred direction, but instead follow a zig-zag motion while approaching each other to fuse.

## Nucleolar response to ATP depletion

To investigate a possible role of active (ATP-dependent) processes in maintaining the nucleolus-nucleoplasm interface, we have examined nucleoli, specifically, their shape, surface roughness and possible fusion events, upon ATP depletion. The ATP was depleted using 2-deoxyglucose and trifluoromethoxy-carbonylcyanide phenylhydrazone (see Materials and methods). *Figure 6A and B* show micrographs of nuclei with fluorescently labeled chromatin (H2B-GFP, green) and nucleoli

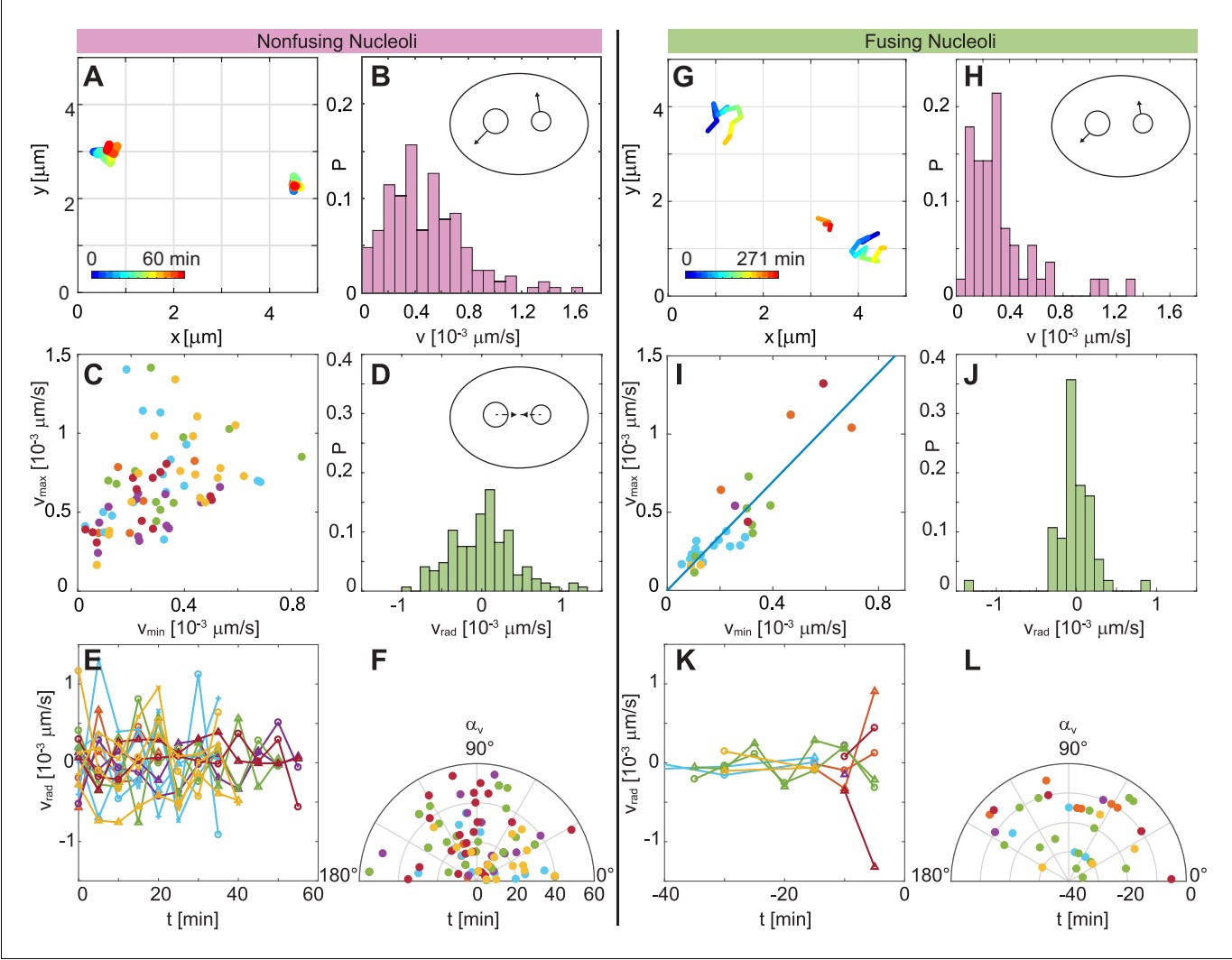

**Figure 5.** Comparison of dynamics between fusing and nonfusing nucleoli. (A) Trajectories of two nonfusing nucleoli color-coded by their temporal evolution (blue to red). The time step is 5 min. (B) Histogram of the velocity magnitude, $v$, for the nonfusing nucleoli ($N_N$ = 17, $N_{cell}$ = 6). (C) Velocities for pairs of nonfusing nucleoli, where $v_{max}$ and $v_{min}$ is the larger and the smaller nucleolar velocity, respectively. (D) Histogram of radial velocity, $v_{rad}$, for nonfusing nucleoli, with $v_{rad}$ calculated with respect to the midpoint distance between nucleoli. (E) $v_{rad}$ as a function of time for nonfusing nucleoli. (F) Velocity angle, $\alpha_v$, in polar coordinates as a function of time for nonfusing nucleoli. (G) Trajectories of pair of fusing nucleoli color-coded by their temporal evolution (blue to red). The pre-fusion nucleoli are visible at earlier times (blue to yellow), while the post-fusion nucleolus appears at later times (orange to red). The time step is 15 and 16 min. (H) Histogram of the $v$ for the fusing nucleoli ($N_N$ = 12, $N_{cell}$ = 7). (I) Velocities for pairs of fusing nucleoli, where $v_{max}$ and $v_{min}$ is the larger and the smaller nucleolar velocity, respectively, with a linear fit $v_{max} \approx 1.74 v_{min}$. (J) Histogram of $v_{rad}$ for fusing nucleoli. (K) $v_{rad}$ as a function of time for fusing nucleoli. (L) $\alpha_v$ for fusing nucleoli as a function of time.

The online version of this article includes the following figure supplement(s) for figure 5:

**Figure supplement 1.** Additional nucleolar trajectories and enlarged view of nucleolar trajectories from *Figure 5A* and *Figure 5G*.

(NPM-DsRed, red) under physiological conditions (control) and upon ATP depletion, respectively. In addition, we review the z-projection of nuclear and nucleolar contours obtained in different planes of the respective z-stack (*Figure 6A–B*). We find that upon ATP depletion nucleoli do not exhibit spherical, but instead irregular shapes (*Figure 6B*). In fact, some of the larger irregularly shaped nucleoli might originate from nucleoli fusing at the time of ATP depletion, given the absence of nucleoli in the hour-glass shape, characteristic of nucleolar fusion under physiological conditions (*Figure 1*).

To characterize the morphological changes of nucleoli upon ATP depletion, we define parameters describing their shape and compare against the control nucleoli. Specifically, after we obtain nuclear

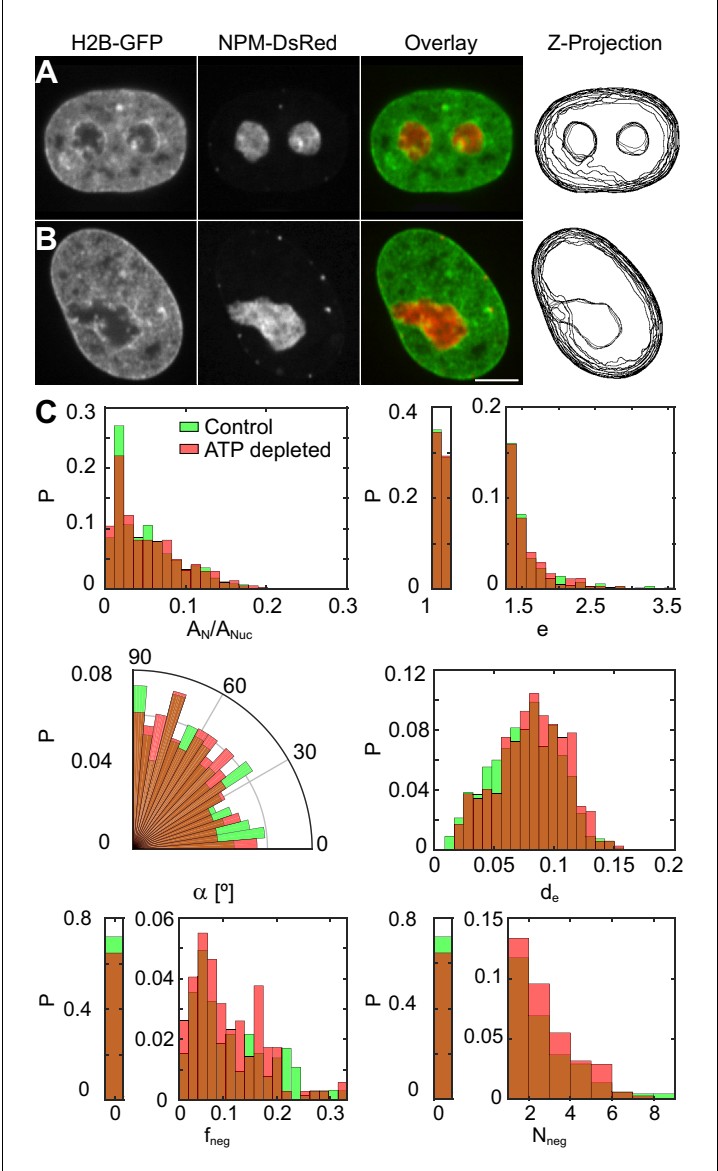

**Figure 6.** Nucleoli under physiological conditions and upon ATP-depletion. (**A–B**) Micrographs of HeLa nuclei with fluorescently labeled chromatin (H2B-GFP, green) and nucleoli (NPM-DsRed, red), their color overlay and z-projections of nucleolar and nuclear contours: (**A**) under physiological conditions (control) and (**B**) after ATP depletion. (**C**) Distributions of the following nucleolar measurements under physiological conditions ($N_N = 648$, $N_{Cell} = 208$) vs. upon ATP depletion ($N_N = 345$, $N_{Cell} = 127$): nucleolar area normalized by nuclear area, $A_N/A_{Nuc}$, nucleolar eccentricity, $e$, angle between the major nuclear and nucleolar axes, $\alpha$, the shortest distance from the nucleolar centroid to the nuclear envelope normalized by the nuclear circumference, $d_e$, fraction of the nucleolar contour with negative curvature, $f_{neg}$, and number of continuous regions of negative curvature along the nucleolar contour, $N_{neg}$. All measurements are carried out in the nucleolar focal plane. Scale bar, 5 μm.

and nucleolar contours in their respective focal planes, we compute the following six parameters for every nucleolus: $A_N/A_{Nuc}$ (the nucleolar area $A_N$ normalized by the nuclear area $A_{Nuc}$), $e$ (the eccentricity $e = a/b$, where $a$ and $b$ are the semi-major and semi-minor nucleolar axis, respectively), $\alpha$ (the angle between the nuclear and nucleolar major axes), $d_e$ (the shortest distance from the nucleolar centroid to the nuclear envelope normalized by the nuclear circumference in the focal plane of the nucleus), $f_{neg}$ (the fraction of the nucleolar contour with negative curvature) and $N_{neg}$ (the number of

independent nucleolar contour regions with negative curvature), where the curvature corresponds to the in plane curvature of the nucleolar contour.

The comparison of these parameters for nucleoli under physiological conditions and upon ATP depletion is shown in *Figure 6C*. *Table 1* provides a summary of means and standard deviations for measured distributions of $A_N/A_{Nuc}$, $e$, $\alpha$, $d_e$, $f_{neg}$ and $N_{neg}$. In addition, we evaluated the $p$-values for all measured physical quantities as well as the relative differences of their means with respect to control (*Table 1*). The relative difference (in %) of the means was calculated as $100 \times [(\mu_Q - \mu_P)/\mu_P]$, where $\mu_P$ is the mean of the probability distribution of the measured physical quantity under control conditions and $\mu_Q$ after the perturbation. Furthermore, we computed the skew of the measured distributions, which informs about their asymmetry, and the Kullback-Leibler divergence with respect to control (*Table 1*). The Kullback-Leibler divergence is a measure of the difference between two probability distributions. It is defined as $\sum P(i) \ln\left(\frac{P(i)}{Q(i)}\right)$, where $P(i)$ and $Q(i)$ are the two distributions. Here, $P(i)$ corresponds to the probability distribution of the measured physical quantity under control conditions, $Q(i)$ after the perturbation.

The most striking change that we observe upon the ATP depletion is the irregularity of the nucleolar shape. The dramatic increase in the nucleolar surface roughness upon ATP depletion is nicely captured by the growing amount of the nucleolar contour possessing negative curvature as quantified by $f_{neg}$ and $N_{neg}$, both showing increase of $\sim$ 20%. Remarkably, when compared to the control nucleoli, the ATP-depleted nucleoli not only show larger parts of their contour to posses negative curvature, but are also more likely to contain several more independent contour regions of negative curvature making them appear lobulated. In addition, the ATP-depleted nucleoli tend to localize further away from the nuclear envelope, as illustrated by $d_e$, than the control nucleoli. Surprisingly, there are no significant changes to the average nucleolar area, eccentricity and orientation as per $A_N/A_{Nuc}$, $e$ and $\alpha$, respectively, upon ATP depletion.

## Nucleolar response to biochemical perturbations

To probe contributions of specific cellular processes (such as cytoskeletal forces, transcriptional activity as well as protein synthesis) to maintaining the nucleolus-nucleoplasm interface, we employ targeted biochemical perturbations. Specifically, to inhibit cytoskeletal forces we treat the cells with blebbistatin, which is a myosin II inhibitor, latrunculin A, which prevents actin polymerization, and nocodazole, which is a microtubule polymerization blocker. To test the contributions of transcription-related processes we apply $\alpha$-amanitin, which inhibits the RNA polymerase II activity, and flavopiridol, which blocks the positive transcription elongation factor P-TEFb. In addition, we probed the impact of the local chromatin packing state by applying trichostatin A, which prevents histone deacetylation, and thus leads to chromatin decondensation. Finally, since the nucleolus is the site of ribosome biogenesis and thus directly involved in cellular protein production, we explore the role of protein synthesis in maintaining the nucleolus-nucleoplasm interface. To do so, we use cycloheximide, a protein synthesis inhibitor, and evaluate its effect at two time points, $t_1 = 30$ min and $t_2 = 6.5$ hr, upon drug addition. We anticipate that at short timescales, we can observe a direct impact of protein synthesis inhibition on the nucleolar-nucleoplasm interface, while at longer timescales, we can investigate a possible feedback between protein synthesis inhibition and nucleolar size and shape.

*Figure 7A* shows micrographs of nuclei with fluorescently labeled chromatin (H2B-GFP, green) and nucleoli (NPM-DsRed, red) under the physiological conditions (control) and after treatment with blebbistatin, latrunculin A, nocodazole, $\alpha$-amanitin, flavopiridol, trichostatin A and cycloheximide at $t_1$ and $t_2$. We also survey the z-projection of nuclear and nucleolar contours obtained in different planes of the respective z-stack (*Figure 7—figure supplement 1*). To examine morphological differences under the studied conditions, we evaluate the same six parameters used earlier: $A_N/A_{Nuc}$, $e$, $\alpha$, $d_e$, $f_{neg}$, and $N_{neg}$ and visualize their distributions as violin plots in *Figure 7B*. The red dot indicates the mean, while the solid and dashed red lines correspond to the median and quartiles, respectively. *Table 1* provides a summary of means and standard deviations for measured distributions of $A_N/A_{Nuc}$, $e$, $\alpha$, $d_e$, $f_{neg}$ and $N_{neg}$. In addition, we evaluated the skew of measured distributions and computed $p$-values for all measured physical quantities, the relative differences of their means with respect to control, as well as the Kullback-Leibler divergence with respect to control (*Table 1*).

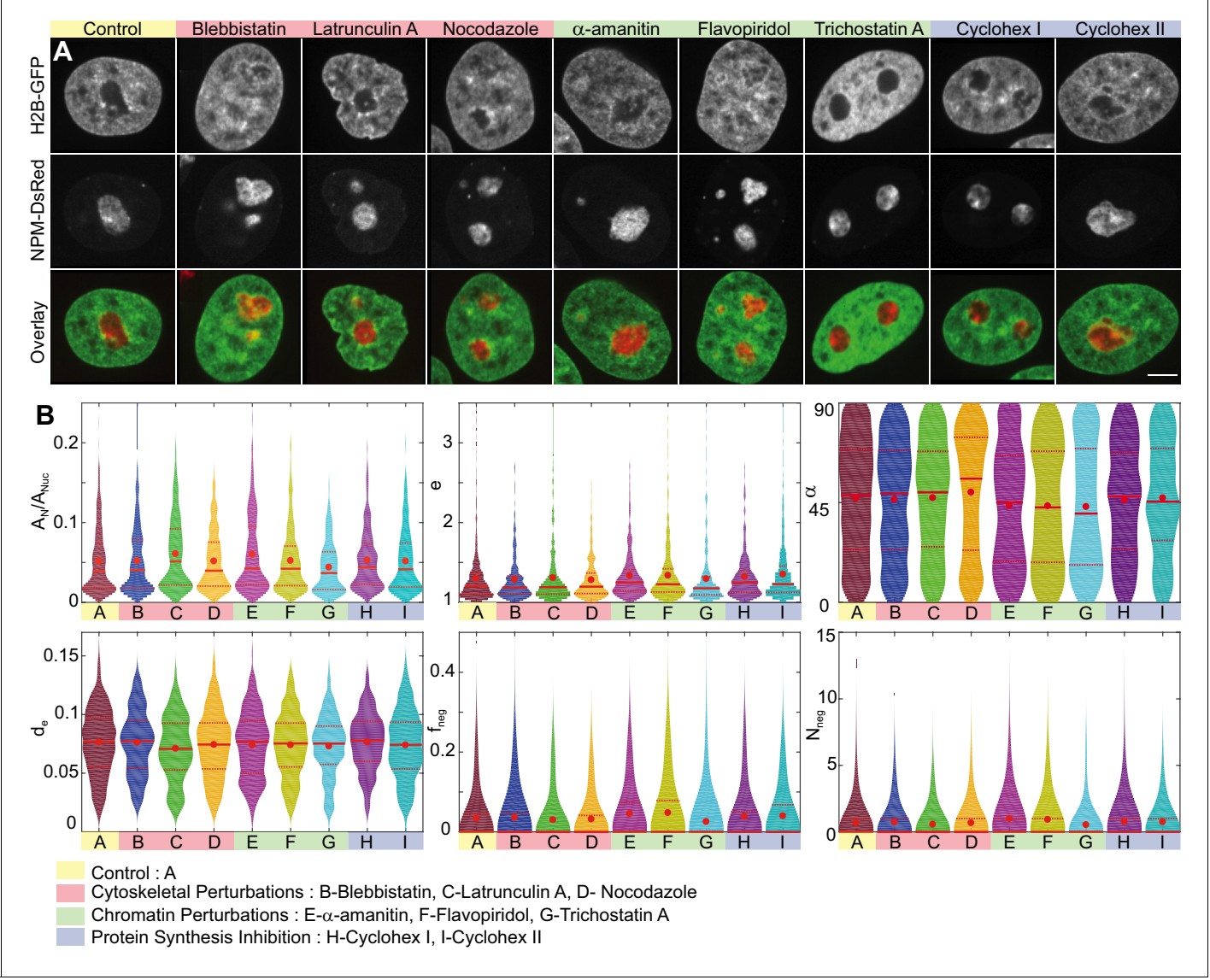

**Figure 7.** Nucleoli upon biochemical perturbations. (**A**) Micrographs of HeLa nuclei with fluorescently labeled chromatin (H2B-GFP, green) and nucleoli (NPM-DsRed, red), and the overlay, under the following conditions: control, upon addition of blebbistatin, latrunculin A, nocodazole, $\alpha$-amanitin, flavopiridol, trichostatin A, and cycloheximide (at $t_1 = 30$ min and $t_2 = 6.5$ hr). (**B**) Histograms of the following measurements for all conditions (width indicates probability): nucleolar area normalized by nuclear area, $A_N/A_{Nuc}$, nucleolar eccentricity, $e$, angle between the major nuclear and nucleolar axes, $\alpha$, the shortest distance from the nucleolar centroid to the nuclear envelope normalized by the nuclear circumference, $d_e$, fraction of the nucleolar contour with negative curvature, $f_{neg}$, and number of continuous regions of negative curvature along the nucleolar contour, $N_{neg}$. All data collected in the nucleolar focal plane. *Red dot*, *solid red line* and *dotted red lines* indicate the mean, median and quartiles, respectively. *Table 1* provides *p*-values for all measured data with respect to the control. The number of nucleoli and cells are as follows: control ($N_N = 648$, $N_{Cell} = 208$), blebbistatin ($N_N = 399$, $N_{Cell} = 127$), latrunculin A ($N_N = 307$, $N_{Cell} = 104$), nocodazole ($N_N = 310$, $N_{Cell} = 106$), $\alpha$-amanitin ($N_N = 268$, $N_{Cell} = 95$), flavopiridol ($N_N = 309$, $N_{Cell} = 105$), trichostatin A ($N_N = 278$, $N_{Cell} = 95$), and cycloheximide at $t_1 = 30$ min ($N_N = 291$, $N_{Cell} = 91$), and cycloheximide at $t_2 = 6.5$ hr ($N_N = 294$, $N_{Cell} = 105$). Scale bar, 5 µm.

The online version of this article includes the following figure supplement(s) for figure 7:

**Figure supplement 1.** Nucleoli upon biochemical perturbations, including z-projections.

**Table 1.** Statistical characteristics of distributions for physical quantities evaluated for nucleoli upon biochemical perturbations (see *Figures 6–7*).

| | $N_{Nucleoli}$ | $N_{Cells}$ | $A_N/A_{Nuc}$ | $e$ | $\alpha$ | $d_e$ | $f_{neg}$ | $N_{neg}$ |
|---|---|---|---|---|---|---|---|---|
| | | | | **Mean $\pm$ standard deviation** | | | | |
| Control | 648 | 208 | 0.052 ± 0.040 | 1.30 ± 0.32 | 47 ± 26 | 0.076 ± 0.029 | 0.033 ± 0.066 | 0.66 ± 1.37 |
| ATP-depletion | 345 | 127 | 0.054 ± 0.042 | 1.29 ± 0.28 | 48 ± 26 | 0.082 ± 0.028 | 0.038 ± 0.064 | 0.82 ± 1.39 |
| Blebbistatin | 399 | 127 | 0.052 ± 0.041 | 1.29 ± 0.28 | 46 ± 26 | 0.076 ± 0.028 | 0.036 ± 0.067 | 0.72 ± 1.38 |
| Latrunculin A | 307 | 104 | 0.061 ± 0.044 | 1.31 ± 0.36 | 47 ± 25 | 0.071 ± 0.028 | 0.030 ± 0.063 | 0.56 ± 1.21 |
| Nocodazole | 310 | 106 | 0.052 ± 0.037 | 1.28 ± 0.25 | 50 ± 28 | 0.074 ± 0.029 | 0.032 ± 0.061 | 0.67 ± 1.38 |
| $\alpha$-amanitin | 268 | 95 | 0.060 ± 0.047 | 1.34 ± 0.29 | 44 ± 27 | 0.074 ± 0.029 | 0.046 ± 0.072 | 0.99 ± 1.71 |
| Flavopiridol | 309 | 105 | 0.052 ± 0.040 | 1.34 ± 0.34 | 43 ± 27 | 0.074 ± 0.028 | 0.048 ± 0.080 | 0.92 ± 1.67 |
| Trichostatin A | 278 | 95 | 0.044 ± 0.032 | 1.29 ± 0.36 | 43 ± 28 | 0.073 ± 0.026 | 0.025 ± 0.063 | 0.53 ± 1.35 |
| Cyclohex I | 291 | 91 | 0.053 ± 0.037 | 1.32 ± 0.29 | 46 ± 26 | 0.076 ± 0.025 | 0.038 ± 0.069 | 0.76 ± 1.49 |
| Cyclohex II | 294 | 105 | 0.052 ± 0.039 | 1.35 ± 0.38 | 47 ± 25 | 0.074 ± 0.029 | 0.040 ± 0.067 | 0.74 ± 1.33 |

| | **p-values (with respect to control)** | | | | | | **Relative difference of mean [%]** | | | | | |
|---|---|---|---|---|---|---|---|---|---|---|---|---|
| | $A_N/A_{Nuc}$ | $e$ | $\alpha$ | $d_e$ | $f_{neg}$ | $N_{neg}$ | $A_N/A_{Nuc}$ | $e$ | $\alpha$ | $d_e$ | $f_{neg}$ | $N_{neg}$ |
| ATP-depletion | 0.359 | 0.674 | 0.513 | 0.009 | 0.258 | 0.093 | 4% | −1% | 2% | 8% | 15% | 24% |
| Blebbistatin | 0.893 | 0.515 | 0.694 | 0.911 | 0.582 | 0.495 | 0% | −1% | −2% | 0% | 9% | 9% |
| Latrunculin A | 0.002 | 0.714 | 0.940 | 0.011 | 0.464 | 0.246 | 17% | 1% | 0% | −7% | −9% | −15% |
| Nocodazole | 0.985 | 0.376 | 0.169 | 0.335 | 0.734 | 0.925 | 0% | −2% | 6% | −3% | −3% | 2% |
| $\alpha$-amanitin | 0.011 | 0.089 | 0.090 | 0.280 | 0.017 | 0.006 | 15% | 3% | −6% | −3% | 39% | 50% |
| Flavopiridol | 0.789 | 0.089 | 0.059 | 0.272 | 0.006 | 0 .018 | 0% | 3% | −9% | −3% | 45% | 39% |
| Trichostatin A | 0.002 | 0.906 | 0.051 | 0.099 | 0.087 | 0.160 | −15% | −1% | −9% | −4% | −24% | −20% |
| Cyclohex I | 0.654 | 0.257 | 0.633 | 0.932 | 0.355 | 0.359 | 2% | 2% | −2% | 0% | 15% | 15% |
| Cyclohex II | 0.984 | 0.035 | 0.982 | 0.284 | 0.163 | 0.381 | 0% | 4% | 0% | −3% | 21% | 12% |

| | **Kullback-Leibler divergence (with respect to control)** | | | | | | **Skew** | | | | | |
|---|---|---|---|---|---|---|---|---|---|---|---|---|
| | $A_N/A_{Nuc}$ | $e$ | $\alpha$ | $d_e$ | $f_{neg}$ | $N_{neg}$ | $A_N/A_{Nuc}$ | $e$ | $\alpha$ | $d_e$ | $f_{neg}$ | $N_{neg}$ |
| Control | – | – | – | – | – | – | 1.10 | 3.07 | −0.09 | −0.04 | 2.17 | 2.68 |
| ATP-depletion | 0.026 | 0.011 | 0.021 | 0.032 | 0.051 | 0.018 | 1.01 | 2.23 | −0.19 | −0.16 | 1.80 | 1.87 |
| Blebbistatin | 0.023 | 0.025 | 0.031 | 0.036 | 0.022 | 0.007 | 1.31 | 2.26 | −0.11 | 0.03 | 2.00 | 2.14 |
| Latrunculin A | 0.059 | 0.029 | 0.028 | 0.040 | 0.050 | 0.015 | 0.79 | 3.01 | −0.15 | −0.07 | 2.06 | 2.36 |
| Nocodazole | 0.022 | 0.014 | 0.042 | 0.025 | 0.033 | 0.010 | 0.87 | 1.84 | −0.28 | 0.07 | 1.90 | 2.55 |
| $\alpha$-amanitin | 0.060 | 0.043 | 0.057 | 0.032 | 0.062 | 0.036 | 0.98 | 1.95 | 0.08 | 0.08 | 1.61 | 2.26 |
| Flavopiridol | 0.038 | 0.027 | 0.044 | 0.035 | 0.053 | 0.021 | 1.28 | 2.36 | 0.03 | −0.02 | 1.62 | 2.09 |
| Trichostatin A | 0.056 | 0.046 | 0.062 | 0.059 | 0.047 | 0.034 | 0.85 | 3.15 | 0.17 | −0.25 | 2.95 | 3.15 |
| Cyclohex I | 0.041 | 0.032 | 0.043 | 0.051 | 0.043 | 0.015 | 1.13 | 2.13 | −0.08 | −0.12 | 1.98 | 2.70 |
| Cyclohex II | 0.013 | 0.026 | 0.049 | 0.021 | 0.040 | 0.014 | 0.98 | 4.19 | −0.08 | 0.01 | 1.66 | 2.14 |

The online version of this article includes the following source data for Table 1:

**Source data 1.** Statistical characteristics of distributions measured in *Figures 6* and *7*.

A close inspection of the violin plots (*Figure 7B*) and their corresponding statistical characteristics (*Table 1*) reveals the following morphological changes upon cytoskeletal, chromatin and protein synthesis perturbations.

Interestingly, the cytoskeletal perturbations, which act on the cytoskeleton outside the cell nucleus, did not lead to any major changes in the nucleolar morphology except for the actin

polymerization inhibitor latrunculin A, which led to an increase of $A_N/A_{Nuc}$ compared to the control nucleoli. This increase, however, is due to the rounding up of nuclei upon the latrunculin A treatment (*Khatau et al., 2009*; *Burnette et al., 2014*), which leads to a decrease in the nuclear area $A_{Nuc}$, and thus causes the apparent increase of $A_N/A_{Nuc}$, while the measured nucleolar area $A_N$ remains comparable to the $A_N$ of control nucleoli. Similarly, the observed decrease in the distance of nucleoli from the nuclear envelope, $d_e$, is likely caused by a decrease of the observed nuclear area $A_{Nuc}$.

In contrast, chromatin perturbations such as transcription inhibitors $\alpha$-amanitin, flavopiridol and histone deacetylase inhibitor trichostatin A led to visible changes in the nucleolar morphology as well as in the roughness of the nucleolus-nucleoplasm interface. Specifically, upon perturbing polymerase II activity using $\alpha$-amanitin, we find that the nucleolar size $A_N/A_{Nuc}$ increases by $\sim 15\%$ and nucleoli are on average more elliptical ($e$). Moreover, the roughness of the nucleolus surface strongly increases by $\sim 40$–$50\%$ as measured by the amount of negative curvature ($f_{neg}$ and $N_{neg}$). Similarly, when we perturb the transcription elongation using flavopiridol, we observe $\sim 40$–$45\%$ increase in the nucleolus surface roughness ($f_{neg}$ and $N_{neg}$) and nucleoli become on average more elliptical ($e$). Finally, when we block the histone deacetylation using trichostatin A, which leads to chromatin decondensation, we find that the nucleolar size $A_N/A_{Nuc}$ decreases by $\sim 15\%$, while their eccentricity ($e$) remains unchanged. Strikingly, the nucleolar surface roughness decreases by $\sim 20$–$25\%$ ($f_{neg}$ and $N_{neg}$), in other words upon trichostatin A treatment it becomes smoother than in the control.

Finally, the protein synthesis inhibition using cycloheximide left the nucleolar size unchanged, while the nucleoli became on average more elliptical ($e$) at longer times ($t_2 = 6.5$ hr). Furthermore, protein synthesis inhibition led to a moderate $\sim 15\%$ increase in the nucleolar surface roughness ($f_{neg}$ and $N_{neg}$) at both times ($t_1 = 30$ min and $t_2 = 6.5$ hr).

The orientation of the nucleoli within the nucleus and the nucleolar distance from the nuclear envelope is not significantly affected by any of the studied perturbations as illustrated by $\alpha$ and $d_e$, respectively.

## Discussion

In this work, we study the nucleolus as the archetype of cellular organelles formed by liquid-liquid phase separation (LLPS) and monitor its size, shape and dynamics during its lifetime in human cells *in vivo*. We discover a rich phenomenology that grows the LLPS framework in new and unexpected ways: (*i*) We find that nucleoli exhibit anomalous dynamics and anomalous volume distribution during the cell cycle that defies any current theory and necessitates a new one. (*ii*) We uncover that the nucleolar fluid is a colloidal solution containing solid-like granules, the DFCs. (*iii*) We reveal that the surrounding nucleoplasm plays a key role in the LLPS of nucleoli that might have been previously overlooked and find that active (ATP-dependent) processes are involved in maintaining the nucleolus-nucleoplasm interface. Moreover, we identify specific biological processes participating in the nucleolus-nucleoplasm interactions.

Our findings show that the nucleolar volume distribution scales as $P(V) \sim V^{-1}$ during the entire cell cycle. The scale-free nature of this distribution suggests that nucleoli of any size can coalesce, moreover, there is no preferred size that nucleoli need to reach before/after they coalesce. It also suggests, that nucleoli of different sizes follow the same coalescence kinetics (*Caragine et al., 2018*). Furthermore, the nucleolar volume distribution remains unchanged during the cell cycle, suggesting that the fusion of nucleoli is not limited to the first two hours of the cell cycle as previously hypothesized (*Savino et al., 2001*), but can occur at any time. Nucleolar coalescence occurs from the early stages in the cell cycle, where it is thought to be a part of the nucleolar assembly process (*Savino et al., 2001*). It is conceivable that at later times, the nucleolar coalescence might serve a different purpose as it is less likely to happen with decreasing nucleolar number.

Interestingly, a volume distribution $P(V) \sim V^{-1}$ was previously found also for liquid-like P-granules in the *C. elegans* oocyte (*Hubstenberger et al., 2013*). In contrast, the volume distribution of nucleoli in the *X. laevis* oocyte follows $\sim V^{-1.5}$, which was shown to be consistent with diffusion-limited aggregation with constant influx of particles (*Brangwynne et al., 2011*). The kinetics of human nucleolar assembly likely differs from that of the frog oocyte due to numerous differences between these two systems. For example, the nucleolar count is much lower in human cells ($\sim 100$ times less than in frog oocyte), there is a dense actin network present in the frog oocyte nucleus (germinal

vesicle), and human somatic nucleoli are connected to the chromatin fiber, thus, they cannot freely diffuse as it is in the case of nucleoli in the *X. laevis* oocyte (*Gall et al., 2004*; *Brangwynne et al., 2011*; *Caragine et al., 2018*; *Berry et al., 2018*). Further differences between the two systems include a large difference in the nuclear size (diameter ~ 1000 μm in frog oocytes, ~ 10 μm in human cells) and the nucleolar size with volumes of 10–$10^3$ μm$^3$ in frog oocytes and $10^{-2}$–$10^2$ μm$^3$ in human cells (*Gall et al., 2004*; *Brangwynne et al., 2011*; *Caragine et al., 2018*).

The anomalous volume distribution of human nucleoli might likely be connected to their anomalous dynamics. Remarkably, our data suggest that one can predict if a pair of nucleoli is going to fuse by analyzing their motion. The differences in the dynamical behavior of non-fusing nucleoli and the ones in approach for fusion are stark. While the non-fusing ones appear to move randomly through the nucleoplasm, nucleoli that will fuse in the near future, move slower than non-fusing ones and show a linear correlation in their velocities (*Figure 5I*). Considering the nucleolar size and the fact that they are physically tethered to the chromatin fiber, their motion unavoidably leads to local spatial reorganization of chromatin. Alternatively, a local chromatin rearrangement could facilitate the nucleolar pre-fusion approach. In fact, in our earlier work we found that the velocities of the growth of the neck connecting two fusing nucleoli (*Figure 1B*) are intriguingly similar to the velocities measured for active chromatin motion (*Caragine et al., 2018*; *Zidovska et al., 2013*). Since nucleoli move in an active fluid (chromatin solution), we speculate that active processes might be involved in bringing them together to undergo fusion. To explore this hypothesis, future experiments and theories are needed to probe the nucleolar interactions with chromatin.

The complex nature of the nucleolar fluid might also contribute to the anomalous behavior of human nuceoli. Strikingly, we find that dense fibrillar components (DFCs) behave as monodisperse solid-like colloidal particles (granules) suspended in a liquid phase of granular component (GC). Our data shows that DFCs do not undergo aggregation, but remain of well-defined size and dimensions with a semi-major axis length of 210 ± 50 nm and semi-minor axis length of 180 ± 40 nm, as well as shape with aspect ratio of 1.22 ± 0.17 even upon nucleolar coalescence, which is consistent with solid-like particles. In contrast, the DFCs in frog oocytes were found to be polydisperse with diameter ~ 2–5 μm, liquid-like with viscoelastic behavior (*Brangwynne et al., 2011*; *Feric et al., 2016*), and with their fusion being observed upon latrunculin A treatment (*Feric et al., 2016*). It is also noteworthy that one frog oocyte DFC can be larger than the entire human nucleolus. Furthermore, our data reveal that human nucleoli obey a volumetric ratio for GC and DFC content, with DFC volume fraction ~ 0.1, which is significantly lower than in frog oocytes (~ 0.25) (*Feric et al., 2016*). This suggests it is the rRNA-rich GC phase that provides the human nucleolus with its liquid-like properties.

To investigate the nucleolar interactions with the surrounding nucleoplasm, we have tested the impact of active (ATP-dependent) processes in general as well as specific biological processes such as cytoskeletal and transcriptional activity, chromatin packing state and protein synthesis. Our data suggest that nucleoli are closely dependent on ATP-dependent processes, losing their spherical shape upon ATP-depletion by exhibiting increased surface roughness (local deformations). In our earlier study (*Caragine et al., 2018*) we have shown that the surface roughness can serve as a readout of the nucleolar surface tension. Specifically, local nucleolar surface deformations, which may be driven thermally or by active processes, are opposed by the surface tension. Thus, the larger the surface roughness, the lower its surface tension. Hence, a possible interpretation of the increase in nucleolar surface roughness upon ATP-depletion is a reduction of the surface tension $\gamma$ of the nucleolus-nucleoplasm interface. These findings are consistent with our earlier study, which found that $\gamma$ under physiological conditions is an effective quantity, and is therefore, likely dependent on some of the ATP-dependent cellular processes (*Caragine et al., 2018*).

Our data show that the roughness (local deformations) of the nucleolus-nucleoplasm interface is highly sensitive to the transcriptional activity in the nucleus. Interestingly, the inhibition of transcriptional activity (such as polymerase II activity or mRNA elongation) in the nucleus leads to an increase of the relative nucleolar size and the nucleolus becomes more elongated (less spherical) with a number of local deformations leading to high surface roughness. However, upon blocking the histone deacetylases, which causes a visible chromatin decondensation (*Tóth et al., 2004*), we find not only a reduction in the relative nucleolar size, but also an increasingly smooth nucleolus-nucleoplasm interface. This suggests that the nucleolus-nucleoplasm interface is closely linked to the chromatin packing state as well as its transcriptional activity. Conversely, the perinucleolar chromatin is mostly heterochromatic, that is largely transcriptionally inactive, yet, its peculiar packing at the nucleolar

surface might require active remodeling. Moreover, this is in agreement with our hypothesis that the surface tension $\gamma$ is maintained by active processes and thus is an effective physical quantity. To elucidate the underlying physics, new theories accounting for the non-equilibrium nature of the nucleolus-nucleoplasm liquid interface need to be developed.

In contrast, we find that the cytoskeletal forces exerted on the nucleus from the cytoplasm, do not contribute to the local roughness of the nucleolus-nucleoplasm interface, nor do they impact the nucleolar size and shape. Interestingly, in frog oocytes the disruption of the dense nuclear actin network by latrunculin A facilitates nucleolar fusion, leading to an increase in the nucleolar size (*Feric et al., 2016*). Conversely, there is no filamentous actin network present in human cell nucleus.

Lastly, our findings reveal that nucleoli are only moderately sensitive to the protein synthesis inhibition at time scales from 30 min to 6.5 hr. We do not observe any change in their size, only a slight increase in the surface roughness. However, it is possible that to observe an effect on nucleoli from the lack of protein synthesis much larger times need to be explored.

In summary, we speculate that the interplay of the complex nature of the nucleolar fluid, the reduced mobility of nucleoli due to their chromatin tethering, as well as their interactions with the surrounding nucleoplasm, might impact the nucleolar assembly kinetics and lead to the observed anomalous nucleolar volume distribution ($\sim V^{-1}$).

In conclusion, nucleoplasm plays a major role in the life of nucleoli, the archetype of the liquid condensate formed by liquid-liquid phase separation in biology. Nucleoplasm, the fluid surrounding the nucleoli, is a complex polymeric solution containing chromatin. Chromatin fiber serves as the template for nucleolar formation and later forms a boundary at the nucleolus-nucleoplasm interface (*McClintock, 1934*; *Ritossa and Spiegelman, 1965*; *Wallace and Birnstiel, 1966*; *Bickmore and van Steensel, 2013*; *Németh and Längst, 2011*; *Towbin et al., 2013*). Strikingly, the DNA sequences at which nucleoli form (NORs) and the genes located at the nucleolar interface are by no means random (*Németh and Längst, 2011*). This likely impacts the 3D chromosomal organization in the nucleolar vicinity. Moreover, considering chromatin's active nature (*Zidovska et al., 2013*) and the fact that nucleoli are tethered to it during their lifetime, we speculate that active positional fluctuations (or rearrangements) of chromatin could bring nucleoli together, facilitating fusion. While this hypothesis remains to be tested, it is consistent with our observations that nucleoli, which are in approach to fusion, exhibit different dynamics than non-fusing ones. It is also supported by previous studies of colloidal mixtures, where the presence of particles with an actively driven translational motion leads to phase separation of active and passive (i.e., thermally driven) components (*Stenhammar et al., 2015*). Similar behavior has been found for polymer mixtures containing active and passive polymers (*Smrek and Kremer, 2017*). We speculate that, with respect to its translational mobility, the nucleolus could be abstracted as a passive droplet (or colloid) immersed in an active polymer (chromatin). In such case, the active positional fluctuations of the polymer could cause demixing of the passive phase and thus effectively bring the passive colloids (nucleoli) together. That is, the active entities phase separate from the passive entities, enabling nucleolar coalescence. Such phase separation is distinct from the liquid-liquid phase separation by which the nucleoli are thought to form at the beginning of the cell cycle (*Brangwynne et al., 2011*; *Berry et al., 2015*; *Feric et al., 2016*).

The nucleolus plays a key role in cellular protein synthesis, thus any changes in nucleolar composition, structure or function can lead to cell abnormalities often connected with human diseases. For example, mutations in nucleolar proteins, which interact with RNA polymerase I, regulate rRNA transcription or participate in rRNA processing, are associated with cell cycle arrest and improper nucleolar assembly. This can lead to diseases such as skeletal and neurodegenerative disorders, cardiovascular disease and cancer (*Hannan et al., 2013*; *Núñez Villacís et al., 2018*; *Ruggero and Pandolfi, 2003*; *Derenzini et al., 2009*). Moreover, in many diseases such as cancer, Alzheimer's and Parkinson's disease, but also in aging, human nucleoli change their shape and size (*Hannan et al., 2013*; *Tsai and Pederson, 2014*; *Núñez Villacís et al., 2018*; *Tiku and Antebi, 2018*), making the nucleolus a potential valuable diagnostic marker. Hence, a mechanistic understanding of nucleolus, its material properties and physical interactions with the nucleoplasm, might illuminate nucleolus in health and disease, contributing to new paths for diagnosis and therapy.

# Materials and methods

**Key resources table**

| Reagent type (species) or resource | Designation | Source or reference | Identifiers | Additional information |
|---|---|---|---|---|
| Cell line (*H. sapiens*) | HeLa | ATCC | ATCC:CCL2 RRID: CVCL_0030 | Stable H2B-GFP cell line |
| Transfected construct (*H. sapiens*) | NPM-DsRed | Addgene | 34553 RRID: Addgene_34553 | |
| Transfected construct (*H. sapiens*) | mCerulean-Fibrillarin-7 | Addgene | 55368 RRID: Addgene_55368 | |
| Transfected construct (*H. sapiens*) | NPM-mApple | *Caragine et al., 2018* | n/a | Generated by Zidovska Lab – published in *Caragine et al., 2018* |
| Chemical compound, drug | RO-3306 | Enzo Life Sciences | ALX-270–463 | |
| Chemical compound, drug | 2-Deoxy-D-Glucose (DOG) | Millipore Sigma | D8375 | |
| Chemical compound, drug | Trifluoromethoxy-carbonylcyanide phenylhydrazone (FCCP) | Millipore Sigma | C2920 | |
| Chemical compound, drug | Latrunculin A | Millipore Sigma | L5163 | |
| Chemical compound, drug | Blebbistatin | Millipore Sigma | B0560 | |
| Chemical compound, drug | Nocodazole | Millipore Sigma | M1404 | |
| Chemical compound, drug | α-Amanitin | Santa Cruz Biotechnology | sc-202440 | |
| Chemical compound, drug | Cycloheximide | Santa Cruz Biotechnology | sc-3508 | |
| Chemical compound, drug | Flavopiridol | Santa Cruz Biotechnology | sc-202157 | |
| Chemical compound, drug | Trichostatin A | Millipore Sigma | T8552 | |
| Software, algorithm | Matlab | MathWorks | 2017a, 2019a | |
| Software, algorithm | Adobe Illustrator, Photoshop | Adobe Inc. | CC2018 | |

## Cell culture and cell transfection

The stable HeLa H2B-GFP cell line was cultured according to ATCC recommendations (CCL-2). Cells were cultured in a humidified, 5% $CO_2$ (vol/vol) atmosphere in Gibco Dulbecco's modified eagle medium (DMEM) supplemented with 10% FBS (vol/vol), 100 units/mL penicillin, 100 µg/mL streptomycin (Invitrogen) and 4.5 µg/mL Plasmocin Prophylactic (Invivogen). Cells were mycoplasma free, as determined by the Invivogen PlasmoTest (Invivogen). For H2B-GFP imaging, cells were plated onto 35 mm MatTek dishes with glass bottom no. 1.5 (MatTek) 24 hr before imaging. We performed four independent experiments. For concurrent H2B-GFP and NPM-DsRed (or NPM-mApple) imaging, cells were plated onto 35 mm MatTek dishes 48 hr before imaging and transiently transfected with NPM-DsRed (or NPM-mApple) 24 hr prior to the experiment. All transfections were carried out using Lipofectamine 2000 (Invitrogen) following the manufacturer's protocol. When indicated, cells were synchronized using 10 µM RO-3306 (Enzo Life Sciences) and imaged before the drug was removed after 16 hr, as well as 3.5 and 5 hr after the drug removal. The synchronized and unsynchronized populations were evaluated in two distinct experiments. For concurrent imaging of H2B GFP, NPM-

DsRed and mCerulean-Fibrillarin-7, cells were plated onto 35 mm MatTek dishes 48 hr before the experiment and transiently transfected with both NPM-DsRed and m-Cerulean-Fibrillarin-7 24 hr prior to the experiment. We performed six independent experiments, all of which were analyzed qualitatively and one quantitatively. NPM-DsRed and FBL-mCerulean (mCerulean3-Fibrillarin-7) were gifts from Mary Dasso (Addgene plasmid # 34553) (*Yun et al., 2008*) and from Michael Davidson (Addgene plasmid # 55368) (*Markwardt et al., 2011*), respectively. NPM-mApple plasmid was created as described earlier (*Caragine et al., 2018*). For experiments involving biochemical perturbations, cells were plated onto 35 mm MatTek dishes 72 hr in advance of the experiment, transiently transfected with NPM-DsRed 48 hr prior to the experiment, and replated onto 35 mm MatTek dishes 24 hr prior to the experiment. We performed three independent experiments for each perturbation and six for the control. All imaging experiments were performed in the Gibco $CO_2$-independent media (Invitrogen) supplemented with L-Glutamine (Invitrogen) and with MatTek dish containing cells mounted on the microscope stage in a custom-built environmental chamber maintained at 37°C with 5% $CO_2$ supplied throughout the experiment.

## Biochemical perturbations

To deplete ATP, cells were treated with 6 mM 2-deoxyglucose (DOG) and 1 μM trifluoromethoxy-carbonylcyanide phenylhydrazone (FCCP) dissolved in $CO_2$-independent medium supplemented with L-glutamine 2 hr before imaging. For cytoskeletal perturbations 10 μM latrunculin A, 10 μM blebbistatin or 10 μM nocodazole, respectively, in $CO_2$-independent medium supplemented with L-glutamine were added to cells 30 min before imaging. For chromatin perturbations, 20 μg/mL α-amanitin (Santa Cruz Biotechnology), 5 μg/mL cycloheximide (Santa Cruz Biotechnology), 83 nM flavopiridol (Santa Cruz Biotechnology), or 624 nM trichostatin A (TSA), respectively, in $CO_2$-independent medium supplemented with L-glutamine were added to cells 30 min, 30 min, 2 hr, and 24 hr before imaging, respectively. For cycloheximide, additional timepoint, 6.5 hr after drug additon, was evaluated. All chemicals were from Sigma Aldrich unless stated otherwise.

## Microscopy and image acquisition

Cells were imaged with a Yokogawa CSU-X1 confocal head with an internal motorized high-speed emission filter wheel, Spectral Applied Research Borealis modification for increased light throughput and illumination homogeneity on a Nikon Ti-E inverted microscope equipped with an oil-immersion 100× Plan Apo NA 1.4 objective lens, an oil-immersion 40× Plan Fluor NA 1.3 objective lens, and the Perfect Focus system. The microscope was mounted on a vibration-isolation air table. The pixel size for the 100× and 40× objective was 0.065 μm and 0.1625 μm, respectively. H2B-GFP and NPM-DsRed (or NPM-mApple) fluorescence was excited with a 488 nm and a 561 nm solid-state laser, respectively. To image H2B-GFP and NPM-DsRed (or NPM-mApple) at the same time, we illuminated the sample simultaneously with both excitation wavelengths, 488 nm and 561 nm. The emission was separated by the W-View Gemini Image Splitter (Hamamatsu) using a dichroic mirror (Chroma Technology), followed by an ET525/30m and an ET630/75m emission filter (Chroma Technology). The two fluorescent signals were allocated to the two halves of the image sensor, producing two distinct images. The exposure time for each frame was 250 ms. For three color imaging, H2B-GFP, NPM-DsRed, and FBL-mCerulean were excited with 488 nm, 561 nm, and 405 nm solid state lasers, respectively, and fluorescence was collected with a 405/488/561/640 multiband-pass dichroic mirror (Semrock) and then an ET525/50m, ET600/50m and ET450/50m emission filter, respectively (Chroma Technology). The exposure time was 250 ms, 250 ms, and 1000 ms for H2B-GFP, NPM-DsRed, and FBL-mCerulean, respectively. Z-stacks were taken with a z axis step size of 500 nm, with the shutter closed in-between steps and an exposure time of 250 ms per plane. Images were obtained with a Hamamatsu ORCA-R2 cooled CCD camera controlled with MetaMorph 7 (Molecular Devices). The streams of 16-bit images were saved as multi-tiff stacks.

## Image processing and data analysis

Images were converted to single-tiff images and analyzed with MatLab (The MathWorks). The nuclear and nucleolar contours were determined from the H2B-GFP and NPM-DsRed signal, respectively, using previously published procedures (*Chu et al., 2017*; *Caragine et al., 2018*).

The nucleolar velocity was determined as the displacement of the centroid of the filled nucleolar contour relative to the displacement of the centroid of the filled nuclear contour, divided by the elapsed time. For radial velocity calculations, we define the radial distance of each nucleolus as its distance from the midpoint of the linear distance between the centroids of two nucleoli. The radial distances for both nucleoli are measured relative to the midpoint of the linear distance between the nucleoli found in two time points, in order to exclude the movement of the other nucleolus in the calculation of nucleolar velocity. Finally, to calculate the radial velocity we divide the change in the radial distance by the time elapsed.

The nucleolar distance from the nuclear envelope was found by finding the minimum distance between the nucleolar centroid and the nuclear contour. The angle between the nucleolus and nucleus was determined by fitting both nucleus and nucleolus with an ellipse and measuring the angle between their respective major axes, for angles greater than 90°, its supplement was taken. The nuclear and nucleolar area were determined as the number of pixels filling its respective contours. Nuclear and nucleolar eccentricity was calculated as the ratio of the semi-major axis length and its semi-minor axis length, when fitted to an ellipse.

To obtain an accurate count of DFCs, we developed a feature-finding procedure. A mask created by the nucleolar contour was applied to the FBL-mCerulean image to remove the background signal. Using a local-maxima function, we found a large number of local maxima indicating possible features in the image, most of which correspond to noise. We manually selected DFCs from the local maxima that were found. Next, we manually fit each DFC with a circumscribed and an inscribed circle, measuring the semi-major DFC axis $a$ and the semi-minor DFC axis $b$, respectively.

## Acknowledgements

This research was supported by the National Institutes of Health Grant R00-GM104152 and by the National Science Foundation Grants CAREER PHY-1554880 and CMMI-1762506.

## Additional information

### Funding

| Funder | Grant reference number | Author |
| --- | --- | --- |
| National Institutes of Health | R00-GM104152 | Alexandra Zidovska |
| National Science Foundation | CAREER PHY-1554880 | Alexandra Zidovska |
| National Science Foundation | CMMI-1762506 | Alexandra Zidovska |

The authors declare that the funders had no role in study design, data collection and interpretation, or the decision to submit the work for publication.

### Author contributions

Christina M Caragine, Resources, Data curation, Software, Formal analysis, Investigation, Visualization, Writing—original draft, Writing—review and editing; Shannon C Haley, Resources, Software, Formal analysis, Investigation; Alexandra Zidovska, Conceptualization, Resources, Data curation, Software, Formal analysis, Supervision, Funding acquisition, Validation, Investigation, Visualization, Methodology, Writing—original draft, Project administration, Writing—review and editing

### Author ORCIDs

Alexandra Zidovska https://orcid.org/0000-0002-4619-4424

### Decision letter and Author response

Decision letter https://doi.org/10.7554/eLife.47533.sa1
Author response https://doi.org/10.7554/eLife.47533.sa2

## Additional files

### Supplementary files
• Transparent reporting form

### Data availability
All data generated or analyzed during this study are included in the manuscript and supporting files.

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
