## [Decision Letter]

Thank you for submitting your article "Nucleolar dynamics and interactions with nucleoplasm in living cells" for consideration by *eLife*. Your article has been reviewed by three peer reviewers, and the evaluation has been overseen by Arup Chakraborty as the Senior and Reviewing Editor. The following individuals involved in review of your submission have agreed to reveal their identity: Rohit V Pappu (Reviewer #1); Pilong Li (Reviewer #3).

The reviewers have discussed the reviews with one another and the Reviewing Editor has drafted this decision to help you prepare a revised submission.

While this paper builds on your past publication in Physical Review Letters in 2018, there are considerable new insights in this manuscript pertaining to the interplay between spontaneous vs. driven processes in regulating nucleolar shapes, sizes, and cell cycle dependent dynamics. However, it is important to highlight the new findings more clearly, and very importantly, to link the data reported in this manuscript to the conclusions drawn. At present, the conclusions are very general and speculative, rather than closely tied to the actual observations. Please find the three reviews appended below as it will help you address additional significant points.

Reviewer #1:

This work is an important and essential follow up to previous studies, one from this very group, on the dynamics of nucleoli. There is growing interest in understanding the interplay between spontaneous and driven (or what some folk refer to as passive vs. active) processes. Yes, facsimiles of nucleoli can be reconstituted in vitro, but the interplay between spontaneous vs. driven processes in regulating nucleolar shapes, sizes, and cell cycle dependent dynamics remains unclear. Here, the authors provide interesting insights based on quantitative imaging that have surprising ramifications. Overall, this is a really important contribution and should be published in *eLife*. Prior to publication there are some issues that need to be addressed and these are listed below.

1) Please clarify whether Figure 1 is a new image or adapted from Caragine et al., 2018.

2) In the fourth paragraph of the Introduction, the authors note that the nucleoplasm is an active fluid and that chromatin dynamics are active and coherent. It would help if the authors were to clarify what they mean by the terms active and coherent. These have specific meanings in the active matter field, and given the possible lack of familiarity that *eLife* readers might have with these concepts. In fact, I'd like to know precisely what the authors mean by active and coherent in this particular context.

3) The authors indicate that "the number of nucleoli per nucleus decreases…"; where is this shown in the data?

4) The fit of the *p(V)* raises some questions of importance: explanation or consideration. In order to be able to interpret the model that emerges from the fits, I propose the following: (1) fit the data to a proper power law of the form *p(V) = f(V)*V^(α – 1)* and fix α to be > 0; (2) then assess the value of α; (3) according to these fits, α should be equal to 2; (4) this would indicate that the distribution of nucleolar volumes follows a model where the moments *V^m* diverge for all values of *m ≥ α – 1*. This implies that estimates of moments, including the first moment viz., the average, will not converge no matter how many samples one extracts. This hypothesis should be tested. I suspect it will be true and if true, the implication is that the underlying features of nucleoli probably derive from Levy flight statistics and are incorrectly analyzed in terms of classical random walks, which lead to erroneous viewpoints such as effective diffusion coefficients and/or effective surface tensions. The underlying asymptotic scale free nature of volume distributions points to interesting properties that aren't explored here, but should perhaps be addressed.

5) Unless I missed it, the sphericity of the DFC wasn't quantitatively analyzed and yet the authors suggest that the DFC exhibits spherical shapes.

6) The trajectories shown in Figure 5A, C, and G of are difficult to parse. Inasmuch as these are trajectories, it might help to have some annotation so one can follow how the trajectories evolve.

7) I felt hard pressed to compare the violin plots to one another. There is a lot of really important information in these plots and the biological implications are rather significant. Therefore, I ask that the authors consider using something like a Kullback-Leibler divergence measure as a way to compare the prior and posterior distributions where the prior would be the default and the posterior would be the effect of the biological perturbation. Absent this, it is hard to judge if the inferences are qualitative or quantitative.

8) The data reveal some extraordinary insights, but interestingly the conclusions seem to either be more general or not about the data themselves. For example, the notion that there are two distinct behaviors for fusing vs. non-fusing nucleoli seems like a bit of a stretch from a quantitative standpoint. Absent a rigorous quantification, I am not convinced that the data do indeed reveal a strong dependence of the nucleolus-nucleoplasm interface on the chromatin transcriptional activity and the packing state. It is an attractive idea, but do the data really support such a strong assertion? I am not entirely sure. As noted in point 4 above, the power law distribution needs further analysis and explanation because this does imply very interesting physics that might even explain the apparent DLCA like behavior that the authors have observed. The authors appear to be invoking a direct connection to ideas from active matter, specifically motility induced phase separation; however, it is not clear that the data directly connect to these ideas. It is more than possible that I have missed the connection. Perhaps the authors would consider elaborating on how their data connect to ideas put forth by Cates and coworkers.

Reviewer #2:

The article by Caragine et al. reports the results of extensive live cell imaging of human HeLa cells expressing H2B-GFP as a chromatin marker and NPM-DsRed, and sometimes also FBL-mCerulean, as nucleolar markers. The senior author, Dr. Zidovska, has previously reported that the high viscosity of the nucleoplasm influences the fluid behavior of nucleoli [Caragine et al., 2018] and extends these studies in the current report to include analysis of nucleolar size and shape at different cell cycle stages, upon depletion of ATP, and after several different biochemical perturbations. The authors emphasize that nucleoli are tethered to and immersed within a fluid sea of chromatin, a perspective often overlook in studies of nucleoli. The results are generally consistent with our understanding of the fluid nature of nucleoli from past studies from others although these authors report the most detailed analysis of human nucleoli to date. The results in Figure 2 show that the size distribution of nucleoli follows a power law with an exponent of -1, in contrast to the exponent of -1.5 observed previously for frog oocyte nucleoli by Cliff Brangwynne. The authors note that differences in the numbers of nucleoli between the two systems may account for the difference in size distribution. Simultaneous observation of NPM1 in the GC and FBL in the DFC showed that DFCs behave like colloidal particles in the fluid GC and that they consistently occupy ~0.1 volume fraction of nucleoli (Figure 3). These results are consistent with findings reported by Cliff Brangwynne in 2016 [Feric, et al., 2016]. The authors compared the movements of nucleoli that, in some cases, lead to encounters and fusion (Figures 4 and 5). They state that the movements of non-fusing and fusing nucleoli are different, as follows: "While the non-fusing ones appear to move randomly through the nucleoplasm, nucleoli that will fuse in the near future, show a bias in motion towards each other." This is a truism; if two nucleoli don't move towards each other, they won't fuse. The authors speculate about the influence of chromatin dynamics on nucleolar fusion but did not perform any experiments to test their ideas. Next, the authors imaged cells after depletion of ATP and observed that nucleoli become irregularly shaped, which they suggested was due to decreased surface tension, although this suggestion was not tested through experiments. Finally, the authors subjected cells to chemical treatments that alter the cytoskeleton, chromatin or protein synthesis and characterized nucleolar number, size and shape using live imaging. In some cases, the treatments led to changes in nucleolar features but this very large dataset did not provide insights of fundamental significance regarding nucleolar structure and function and interactions with the surrounding chromatin. The Conclusions section of the paper presents a series of almost exclusively speculative statements about various aspects of interactions between fluid chromatin and nucleoli. Many of these are interesting hypotheses for testing in the future but are not appropriate as conclusion statements in a primary research article in *eLife*.

The article presents the results of a powerful live cell imaging platform capable of monitoring the structural and dynamic features of chromatin and nucleoli in live cells in real time. However, the results presented largely confirm past observations regarding the features of nucleoli and do not provide new insights. The results can serve as a starting point for future investigations of, for example, links between the state of surrounding chromatin and nucleolar function, which the authors are encouraged to undertake.

Reviewer #3:

The authors extensively studied the dynamics and morphology of individual nucleoli as well as relative dynamics between fusing or non-fusing nucleoli pairs in HeLa cells. These experiments are further extensions of their recent Physical Review Letters paper. However, I find their current effort compelling as they are inferring nucleoli-nucleoplasm interactions, an understudied important topic in the field of LLPS. The authors have established a number of quantitative measurements. Using these assays, they were able to evaluate the effects of ATP hydrolysis on dynamics of nucleoli and have found out some ATP-dependent cellular processes likely regulate dynamics of nucleoli. They also evaluated alterations to dynamics of nucleoli by perturbations to cytoskeleton dynamics, transcription, epigenetic regulation, or protein synthesis. For example, they have found out that alterations to chromatin compaction via epigenetic perturbation cause salient changes in nucleoli dynamics. While it is well-accepted that ATP-dependent processes can regulate dynamics biomolecular condensates, the observation of potential interplay between global epigenetics landscape and nucleoli regulation is rather intriguing. It is highly conceivable that scientists in the field can reveal a lot more nucleoli-nucleoplasm interactions and other biomolecular condensates-surrounding environment interactions by combining these quantitative measurements and genetic perturbations. For at least these reasons, I recommend the publication of this manuscript in *eLife*.

---

## [Author Response]

While this paper builds on your past publication in PRL in 2018, there are considerable new insights in this manuscript pertaining to the interplay between spontaneous vs. driven processes in regulating nucleolar shapes, sizes, and cell cycle dependent dynamics. However, it is important to highlight the new findings more clearly, and very importantly, to link the data reported in this manuscript to the conclusions drawn. At present, the conclusions are very general and speculative, rather than closely tied to the actual observations. Please find the three reviews appended below as it will help you address additional significant points.

We thank the editor for the valuable feedback and the three reviews. In our original manuscript, we have included our data-related conclusions interspersed throughout the Results section and listed only general conclusions in the Conclusions section, which we now realize must have been confusing. Encouraged by the editor’s and reviewers’ comments, we have remedied this in the revised manuscript, by listing and explaining all our data-related conclusions in a new section Discussion and Conclusions. We have also introduced a new Table 1, which provides an overview of our quantitative results. We hope this will help to clarify many of the questions below.

Reviewer #1:[…]1) Please clarify whether Figure 1 is a new image or adapted from Caragine et al., 2018.

We thank the reviewer for pointing this out. Images in Figure 1A are new and were not previously published, images in Figure 1B are in part new and in part adapted from Caragine et al., 2018; specifically, only 2 panels from 14 panels in Figure 1B were previously shown and have been adapted for *eLife*. To account for this, in the revised manuscript we added an explanatory note into the Figure 1 legend.

2) In the fourth paragraph of the Introduction, the authors note that the nucleoplasm is an active fluid and that chromatin dynamics are active and coherent. It would help if the authors were to clarify what they mean by the terms active and coherent. These have specific meanings in the active matter field, and given the possible lack of familiarity that eLife readers might have with these concepts. In fact, I'd like to know precisely what the authors mean by active and coherent in this particular context.

The reviewer raises a great point and we completely agree that the terms “active” and “coherent” referring to the chromatin dynamics should be explained. By “active”, we mean that the chromatin dynamics is ATP-dependent, i.e. it dissipates energy [Zidovska et al., 2013]. In the same study we found that chromatin dynamics is “coherent” over several seconds and microns, i.e. chromatin motion is correlated within micron-scale patches inside the cell nucleus [Zidovska et al., 2013]. In the revised manuscript, we have added an explanation for both terms in the fourth paragraph of the Introduction.

3) The authors indicate that "the number of nucleoli per nucleus decreases…"; where is this shown in the data?

We thank the reviewer for pointing this out. The fact that the number of nucleoli decreases with the progressing cell cycle is shown in Figure 2E, specifically, in the temporal evolution of the distributions measured at 1.5h (blue), 3h (yellow) and G2/M (green). The distribution becomes heavier at small nucleolar numbers (1-5) as the time progresses indicating that more nuclei possess smaller number of nucleoli.

4) The fit of the p(V) raises some questions of importance: explanation or consideration. In order to be able to interpret the model that emerges from the fits, I propose the following: (1) fit the data to a proper power law of the form p(V) = f(V)*V^(α – 1) and fix α to be > 0; (2) then assess the value of α; (3) according to these fits, α should be equal to 2; (4) this would indicate that the distribution of nucleolar volumes follows a model where the moments V^m diverge for all values of m ≥ α – 1. This implies that estimates of moments, including the first moment viz., the average, will not converge no matter how many samples one extracts. This hypothesis should be tested. I suspect it will be true and if true, the implication is that the underlying features of nucleoli probably derive from Levy flight statistics and are incorrectly analyzed in terms of classical random walks, which lead to erroneous viewpoints such as effective diffusion coefficients and/or effective surface tensions. The underlying asymptotic scale free nature of volume distributions points to interesting properties that aren't explored here, but should perhaps be addressed.

The reviewer raises a great point. The fact that the nucleolar volume distribution scales as *p(V) ~ V^-1^*has very interesting consequences. Indeed, such distribution is divergent and so is its first moment, the mean. To examine this property of our *p(V)* distribution, we have performed the following statistical test: for each condition from Figure 2F, e.g. unsynchronized cells (*N_N_* = 1331), we have split our data into 3 random nonoverlapping subpopulations and evaluated their means. We find that the means are quite similar. This is not unexpected, as our *p(V)* distribution does have finite bounds, specifically, there are physical cut-offs as to what size a nucleolus can have: with maximum being the size of the cell nucleus and minimum being the molecular scale of nucleolar components, although in practice the minimum size that can be experimentally measured, is given by the optical resolution. Thus, in between those bounds, *p(V)* can be integrated and also possesses a mean. However, the scale-free nature of *p(V)* even in between those bounds suggests that there is no preferred length scale associated with the nucleolar size. This is very interesting as it suggests that nucleoli of any size can coalesce and there is no preferred size they need to reach before/after they coalesce.

In the revised manuscript we have expanded the discussion of the intriguing properties and consequences of the measured volume distribution in the Results section and in the Discussion and Conclusions section. To more fully investigate the mechanism underlying this phenomenon we propose to address it in a future study. We also hope that our experimental observations will encourage theorists to develop new theories describing this phenomenon.

5) Unless I missed it, the sphericity of the DFC wasn't quantitatively analyzed and yet the authors suggest that the DFC exhibits spherical shapes.

The reviewer is correct, in the original manuscript we have not measured the sphericity of DFCs, since a visual inspection of our data revealed that DFCs appear to be close to spherical. Thus, we fitted the observed DFCs with circles to extract their areas/radii. Encouraged by the reviewer’s comment, we have developed a new data analysis that would allow us to assess the eccentricity of each DFC. To do so, we have developed a feature finding algorithm to identify DFCs, which we then manually verify and fit each DFC with a circumscribed and an inscribed circle, measuring the semi-major DFC axis *a* and the semi-minor DFC axis *b,* respectively. This way, we can evaluate the DFC eccentricity *e=a/b*. Using this algorithm we identified 1279 DFCs over 114 nucleoli in 63 nuclei, and after the removal of the DFCs that were out of focus, we obtained measurements of *a, b, e* and area for 1035 DFCs.

Our results are shown in the revised Figure 3B, where we present the measurements of both the semi-major DFC axis *a* (red) and the semi-minor DFC axis *b* (green), together with the Gaussian fits of their respective distributions. From the Gaussian fits we obtain the following average values: *a_DFC_ =* 210 ± 50 nm and *b_DFC_ =* 180 ± 40 nm.

Next, we evaluate the eccentricity *e=a/b* for each DFC. The distribution of the measured eccentricities *e* is shown in Figure 3—figure supplement 1A. Our data shows, that the DFC shape is indeed very close to spherical with the mean *e =* 1.22 ± 0.17. Figure 3—figure supplement 1B shows the distribution of areas of single DFCs, where the DFC area is calculated as *A =* π*ab* for every DFC. Figure 3—figure supplement 1C shows the distribution of volumes for the DFCs, where *V=4*π*a^3^/3*, providing the upper boundary on the volume estimate.

In the revised manuscript, we have inserted the description of the DFC eccentricity measurement in the Results section, and the Materials and Methods section. We have also updated Figure 3B and C with the plots of data using our new algorithm and revised Figure 3—figure supplement 1.

6) The trajectories shown in Figure 5A, C, and G of are difficult to parse. Inasmuch as these are trajectories, it might help to have some annotation so one can follow how the trajectories evolve.

We thank the reviewer for pointing this out. To remedy this, we have replaced the groups of trajectories originally shown in Figure 5A and G with a single set of trajectories giving an example of a non-fusing and fusing pair of nucleoli. This way the trajectory plots are much larger and easier to parse. In addition, we have color-coded these trajectories by their time progression (blue to red), visualizing their evolution in time. More examples of trajectories with color-coded time progression, for both non-fusing and fusing nucleoli, are presented in the Figure 5—figure supplement 1.

7) I felt hard pressed to compare the violin plots to one another. There is a lot of really important information in these plots and the biological implications are rather significant. Therefore, I ask that the authors consider using something like a Kullback-Leibler divergence measure as a way to compare the prior and posterior distributions where the prior would be the default and the posterior would be the effect of the biological perturbation. Absent this, it is hard to judge if the inferences are qualitative or quantitative.

We thank the reviewer for the helpful suggestion. We agree that a quantitative comparison of the distributions of the physical quantities would be very helpful. To that end we have expanded our analysis of the distribution statistics also to the comparative analysis. Thus, in addition to the means and standard deviations of all distributions and *p*-values with respect to the control (Table 1), in the revised manuscript, we have also computed the relative difference of means with respect to control, the skew of all distributions and the Kullback-Leibler divergence with respect to the control (Table 1). Such analysis allows us to compare the measured distributions against the control as well as to evaluate the statistical significance of their deviation from the control, showing that some of the observed changes are indeed significant. We present all statistical characteristics that we computed for the measured distributions in Table 1 in the revised manuscript and provide their definitions and discuss the specific significant changes in the Results section.

8) The data reveal some extraordinary insights, but interestingly the conclusions seem to either be more general or not about the data themselves.

We thank the reviewer for this helpful comment and apologize for the confusion. In our original manuscript, we have included our data-related conclusions interspersed throughout the Results section and listed only general conclusions in the Conclusions section, which we now realize must have been confusing. Encouraged by the reviewer’s comment, we have remedied this in the revised manuscript, by listing and explaining all our data-related conclusions in a new section Discussion and Conclusions. This might clarify many of the questions below.

For example, the notion that there are two distinct behaviors for fusing vs. non-fusing nucleoli seems like a bit of a stretch from a quantitative standpoint.

The reviewer raises a valid point. We based our claim that fusing and non-fusing nucleoli exhibit different dynamics on purely quantitative evidence analyzing the dynamic behavior of fusing and non-fusing nucleoli (Figure 5). As shown in Figure 5C and I, the velocity amplitudes of two fusing nucleoli are highly correlated with each other exhibiting strong linear dependence (Figure 5I, Pearson correlation coefficient ρ = 0.88), unlike the non-fusing nucleoli, whose velocity amplitudes are totally independent (Figure 5C, Pearson correlation coefficient ρ = 0.41). Further, the velocities measured for fusing nucleoli are 30% smaller than for non-fusing nucleoli (Figure 5B and H), with *p*-value of 4⋅10^-4^. Similarly, the distribution of radial velocities exhibit much larger variance for non-fusing nucleoli than for the fusing ones (Figure 5D and J), with the variance obtained by fitting a Gaussian curve to both distributions: σ*_rad, fusing_* = 1⋅10^-4^ µms^-1^ for the fusing nucleoli and σ*_rad, non-fusing_* = 3⋅10^-4^ µms^-1^ for the non-fusing nucleoli. All of these differences are statistically significant and hint at a possible process responsible for bringing the fusing nucleoli together.

In the revised manuscript we address this point in the Results and the new Discussion and Conclusions sections. We plan to investigate further the mechanism beyond this phenomenon in a future study.

Absent a rigorous quantification, I am not convinced that the data do indeed reveal a strong dependence of the nucleolus-nucleoplasm interface on the chromatin transcriptional activity and the packing state. It is an attractive idea, but do the data really support such a strong assertion? I am not entirely sure.

We thank the reviewer for raising this point and apologize for the confusion. Our data shows that the nucleolus-nucleoplasm interface is highly sensitive to chromatin transcriptional activity and packing state, the former evidenced by large (40-50%) increase in the roughness of the nucleolus-nucleoplasm interface (*f_neg_, N_neg_*) upon α-amanitin and flavopiridol treatment (both transcription inhibitors) and (~15%) increase in nucleolar size upon α-amanitin treatment, the latter evidenced by a significant (~20%) decrease in the roughness of the nucleolus-nucleoplasm interface and (~15%) decrease in nucleolar size upon trichostatin A treatment (leads to chromatin decondensation). Specifically, α-amanitin is a polymerase II transcription inhibitor, flavopiridol is a transcription elongation inhibitor and trichostatin A is a histone deacetylase inhibitor leading to chromatin decondensation. The statistical significance of all the observed changes was verified by evaluating the *p*-values and as per reviewer’s suggestion we also computed the Kullback-Leibler divergences for measured distributions of physical quantities. We have also evaluated the skew and the relative changes of means measured for distributions of physical quantities upon biochemical perturbations with respect to the control.

To strengthen this point, in the revised manuscript, we have introduced Table 1 in Results section, which provides the summary of statistical characteristics for measured distributions of physical quantities upon biochemical perturbations such as mean, standard deviations, *p*-values (with respect to control), the relative differences of means (with respect to control), skew and the Kullback-Leibler divergence (with respect to control). Table 1 provides a direct overview of the observed quantitative changes. Furthermore, we also emphasize these results in the Results section and in the new Discussion and Conclusions section.

As noted in point 4 above, the power law distribution needs further analysis and explanation because this does imply very interesting physics that might even explain the apparent DLCA like behavior that the authors have observed.

See point 4 above.

The authors appear to be invoking a direct connection to ideas from active matter, specifically motility induced phase separation; however, it is not clear that the data directly connect to these ideas. It is more than possible that I have missed the connection. Perhaps the authors would consider elaborating on how their data connect to ideas put forth by Cates and coworkers.

We thank the reviewer for bringing up this point. In earlier studies [Stenhammer et al., 2015, Smrek and Kremer, 2017], an activity driven phase separation was observed in both polymer and colloidal systems consisting of a mixture of active and passive components. In both systems (polymers, colloids), the active entities phase separate from the passive ones. We speculate that in our case the nucleolus could be abstracted as a passive droplet (or colloid) immersed in the active polymer (chromatin). In such case, it is conceivable that the active positional fluctuations of the polymer (chromatin) could cause demixing of the passive phase and thus bring together the passive colloids (nucleoli), i.e. the active entities phase separate from the passive entities in the system. As the mechanism that brings two nucleoli remains mysterious, we speculate that the active motion of the chromatin could indeed contribute to bringing the nucleoli together. In the revised manuscript, we expand our discussion of this point in the new Discussion and Conclusions section.

Reviewer #2:[…] The results in Figure 2 show that the size distribution of nucleoli follows a power law with an exponent of -1, in contrast to the exponent of -1.5 observed previously for frog oocyte nucleoli by Cliff Brangwynne. The authors note that differences in the numbers of nucleoli between the two systems may account for the difference in size distribution.

We thank the reviewer for raising this point. We note that the kinetics of human nucleolar assembly likely differs from that of the frog oocyte due to the much lower nucleolar count in human cells (~100 times less than in frog oocyte), as well as the fact that human somatic nucleoli are connected to the chromatin fiber and therefore are not free to diffuse as it is in the case of nucleoli in the *X. laevis* oocyte [Gall et al., 2004, Brangwynne et al., 2011, Caragine et al., 2018, Berry et al., 2018]. Further differences between the two systems include a large difference in the nuclear size (~ 1 mm diameter in frog oocytes, ~ 10 µm diameter in human cells) and the nucleolar size with volumes of 10–10^3^ µm^3^ in frog oocytes [Brangwynne et al., 2011] and 10^-2^–10^2^ µm^3^ in human cells [current study]. Moreover, frog oocytes contain a dense actin network, which was also implicated in the maintenance of the nucleolar interactions [Feric et al., 2016]. Considering all of these differences, it is conceivable that the different nucleolar volume distributions observed for frog oocytes, *p(V)~V^-1.5^*, and human cells, *p(V)~V^-1^*, might hint at two distinct mechanisms at play in the two systems.

In the revised manuscript, we have extended our discussion of this point in the Discussion and Conclusions section.

Simultaneous observation of NPM1 in the GC and FBL in the DFC showed that DFCs behave like colloidal particles in the fluid GC and that they consistently occupy ~0.1 volume fraction of nucleoli (Figure 3). These results are consistent with findings reported by Cliff Brangwynne in 2016 [Feric, et al., 2016].

The reviewer raises a good point. While our DFC data is in general consistent with the earlier study by [Feric et al., 2016], it also shows striking differences between the two studied systems: frog oocytes in [Feric et al., 2016] and human cells in this work. Specifically, the DFCs in frog oocytes were found to behave as viscoelastic droplets of varying sizes in micrometer range, which can fuse [Feric et al., 2016]. In contrast, we find that DFCs in human cells behave as solid-like colloidal particles, which do not fuse. Moreover, these colloids are monodisperse granules of submicron size with very well defined dimensions: semi-major axis 210 ± 50 nm, semi-minor axis 180 ± 40 nm and eccentricity ~ 1.2 ± 0.17. Furthermore, the DFC volume fraction of nucleoli is only ~ 0.1 human cells, while it is ~ 0.25 in frog oocytes [Feric et al., 2016]. It is also noteworthy that one frog oocyte DFC can be larger than the entire human nucleolus. Thus, the overall differences between the frog oocyte and human cells are indeed significant.

In the revised manuscript we have addressed this point in the Discussion and Conclusions section. We have also inserted new measurements of the DFC semi-major and semi-minor axes (Figure 3B), as well as eccentricity (Figure 3—figure supplement 1A) in the Results section.

The authors compared the movements of nucleoli that, in some cases, lead to encounters and fusion (Figures 4 and 5). They state that the movements of non-fusing and fusing nucleoli are different, as follows: "While the non-fusing ones appear to move randomly through the nucleoplasm, nucleoli that will fuse in the near future, show a bias in motion towards each other." This is a truism; if two nucleoli don't move towards each other, they won't fuse.

We thank the reviewer for pointing this out and apologize for the confusion. What we intended to state was that two fusing nucleoli exhibit a linear correlation in the magnitudes of their velocities (Figure 5I). Conversely, the velocity magnitudes of non-fusing nucleoli are uncorrelated (Figure 5C). In addition, the velocity magnitudes, both absolute (Figure 5B and H) and radial (Figure 5D and J), of fusing nucleoli are on average smaller than those of non-fusing ones. All of these differences are statistically significant as verified by the Pearson coefficient and *p-*values. In the revised manuscript, we have rephrased the statement in the subsection “Dynamics of Fusing and Non-Fusing Nucleoli”.

The authors speculate about the influence of chromatin dynamics on nucleolar fusion but did not perform any experiments to test their ideas.

We thank the reviewer for bringing this up. This comment was meant to be speculative and thought-provoking following up on our data-related conclusions. We realize that it might have been confusing as in the original manuscript we had our data-related conclusions interspersed throughout the Results section. In the revised manuscript, we have summarized all our data-related discussion and conclusions in a new section Discussion and Conclusions.

Next, the authors imaged cells after depletion of ATP and observed that nucleoli become irregularly shaped, which they suggested was due to decreased surface tension, although this suggestion was not tested through experiments.

We thank the reviewer for raising this point. In our earlier study [Caragine et al., 2018] we have established a direct connection between the surface roughness and surface tension of the nucleoli. Specifically, local nucleolar surface deformations, which may be driven thermally or by active processes, are opposed by the surface tension. The surface tension drives the minimization of the surface of a liquid-liquid interface, i.e. a 3D liquid droplet strives to become a sphere. Thus, the larger the surface roughness, i.e. the less spherical the droplet, the lower its surface tension. Under this assumption, the effective surface tension of the nucleoli upon ATP depletion is lower than in the control, since we found that nucleoli exhibit rougher (more uneven) surface as quantified by the fraction of negative curvature *f_neg_* and the total number of the regions of negative curvature *N_neg_.* These experiments and findings are summarized in Figure 6 and a newly introduced Table 1. While we are not measuring the absolute value of the surface tension, our experiments allow for relative comparison between the control and ATP-depleted nucleoli via *e, f_neg_*and *N_neg_*. In the revised manuscript, we address this point in detail in the Discussion and Conclusions section.

Finally, the authors subjected cells to chemical treatments that alter the cytoskeleton, chromatin or protein synthesis and characterized nucleolar number, size and shape using live imaging. In some cases, the treatments led to changes in nucleolar features but this very large dataset did not provide insights of fundamental significance regarding nucleolar structure and function and interactions with the surrounding chromatin.

We thank the reviewer for raising this point and apologize for the confusion. The main focus of our large dataset was to interrogate the nucleoplasm-nucleolus interactions, specifically, involvement of certain biological processes. To do so, we have evaluated in great detail the shape and roughness of the nucleoplasm-nucleolus interface upon biochemical perturbations perturbing specific processes. Our data shows that the nucleolus-nucleoplasm interface is highly sensitive to chromatin transcriptional activity and packing state, the former evidenced by large (40-50%) increase in the roughness of the nucleolus-nucleoplasm interface (*f_neg_, N_neg_*) upon α-amanitin and flavopiridol treatment (both transcription inhibitors) and (~15%) increase in nucleolar size upon α-amanitin treatment, the latter evidenced by a significant (~20%) decrease in the roughness of the nucleolus-nucleoplasm interface and (~15%) decrease in nucleolar size upon trichostatin A treatment (leads to chromatin decondensation). Specifically, α-amanitin is a polymerase II transcription inhibitor, flavopiridol is a transcription elongation inhibitor and trichostatin A is a histone deacetylase inhibitor leading to chromatin decondensation. The statistical significance of all these changes was verified by evaluating the *p*-values as well as the Kullback-Leibler divergences for measured distributions of physical quantities. Our findings suggest that the nucleolus-nucleoplasm interface is dependent on certain active processes, and thus extends our equilibrium liquid-liquid phase separation (LLPS) picture of the nucleolus.

Furthermore, we find that cytoskeletal perturbations do not lead to any significant changes in the nucleolar size as well as the roughness of the nucleolus-nucleoplasm interface (*f_neg_, N_neg_*). Conversely, in frog oocytes latrunculin A facilitates nucleolar fusion as the dense nuclear actin network dissolves leading to an increase in the nucleolar size [Feric et al., 2016].

In the revised manuscript, we have introduced Table 1 in Results section, which provides the summary of statistical characteristics for measured distributions of physical quantities upon biochemical perturbations such as mean, standard deviations, *p*-values (with respect to control), skew and the Kullback-Leibler divergence (with respect to control). Table 1 provides a direct overview of the observed quantitative changes. Furthermore, we also emphasize these results in the Results section, and in the new Discussion and Conclusions section.

The Conclusions section of the paper presents a series of almost exclusively speculative statements about various aspects of interactions between fluid chromatin and nucleoli. Many of these are interesting hypotheses for testing in the future but are not appropriate as conclusion statements in a primary research article in eLife.

We thank the reviewer for this helpful comment and apologize for the confusion. In our original manuscript, we have included our data-related conclusions interspersed throughout the Results section and listed only general conclusions in the Conclusions section, which we now realize must have been confusing. Encouraged by the reviewer’s comment, we have remedied this in the revised manuscript, by listing and explaining all our data-related conclusions in a new section Discussion and Conclusions.

The article presents the results of a powerful live cell imaging platform capable of monitoring the structural and dynamic features of chromatin and nucleoli in live cells in real time. However, the results presented largely confirm past observations regarding the features of nucleoli and do not provide new insights. The results can serve as a starting point for future investigations of, for example, links between the state of surrounding chromatin and nucleolar function, which the authors are encouraged to undertake.

We thank the reviewer for feedback and valuable comments. In our original manuscript, we have included our data-related conclusions interspersed throughout the Results section and listed only general conclusions in the Conclusions section, which we now realize must have been confusing. Encouraged by the reviewer’s comments, we have remedied this in the revised manuscript, by listing and explaining all our data-related conclusions in a new section Discussion and Conclusions. This might clarify many of the questions above.

In this work, we study the nucleolus as the archetype of the liquid-liquid phase separation (LLPS) in biology and monitor its size, shape and dynamics during its entire lifetime. We discover a rich phenomenology that grows the LLPS framework in new and unexpected ways:

i) We find that nucleoli exhibit anomalous dynamics and anomalous volume distribution during the cell cycle that defies any current theory and necessitates a new one.

ii) We find that the nucleolar fluid in human cells is in fact a colloidal solution.

iii) We find that the surrounding nucleoplasm plays a key role in the LLPS of nucleoli that might have been previously overlooked.

iv) We identify specific biological processes participating in the nucleolus-nucleoplasm interactions.